# Purinergic GPCR-integrin interactions drive pancreatic cancer cell invasion

Elena Tomas Bort[1,2], Megan D Joseph[3,4], Qiaoying Wang[1], Edward P Carter[1], Nicolas J Roth[1,2], Jessica Gibson[1], Ariana Samadi[1], Hemant M Kocher[1], Sabrina Simoncelli[3,4], Peter J McCormick[2]*, Richard P Grose[1]*

[1]Centre for Tumour Biology, Barts Cancer Institute, Queen Mary University of London, London, United Kingdom; [2]Centre for Endocrinology, William Harvey Research Institute, Queen Mary University of London, London, United Kingdom; [3]London Centre for Nanotechnology, University College London, London, United Kingdom; [4]Department of Chemistry, University College London, London, United Kingdom

*For correspondence:
p.mccormick@qmul.ac.uk
(PJMcC);
r.p.grose@qmul.ac.uk (RPG)

Competing interest: The authors declare that no competing interests exist.

**Abstract** Pancreatic ductal adenocarcinoma (PDAC) continues to show no improvement in survival rates. One aspect of PDAC is elevated ATP levels, pointing to the purinergic axis as a potential attractive therapeutic target. Mediated in part by highly druggable extracellular proteins, this axis plays essential roles in fibrosis, inflammation response, and immune function. Analyzing the main members of the PDAC extracellular purinome using publicly available databases discerned which members may impact patient survival. *P2RY2* presents as the purinergic gene with the strongest association with hypoxia, the highest cancer cell-specific expression, and the strongest impact on overall survival. Invasion assays using a 3D spheroid model revealed P2Y$_2$ to be critical in facilitating invasion driven by extracellular ATP. Using genetic modification and pharmacological strategies, we demonstrate mechanistically that this ATP-driven invasion requires direct protein-protein interactions between P2Y$_2$ and αV integrins. DNA-PAINT super-resolution fluorescence microscopy reveals that P2Y$_2$ regulates the amount and distribution of integrin αV in the plasma membrane. Moreover, receptor-integrin interactions were required for effective downstream signaling, leading to cancer cell invasion. This work elucidates a novel GPCR-integrin interaction in cancer invasion, highlighting its potential for therapeutic targeting.

## Editor's evaluation

In this manuscript, the authors address an important and urgent question: what molecular mechanisms drive the invasive behavior of pancreatic adenocarcinoma? Because these tumors have such a strong propensity for invasion and metastasis, identifying actionable targets is of high importance. Using a combination of in silico and in vitro modeling, they identify a role for purinergic G-protein coupled receptor P2Y$_2$ as a critical node in mediating PDAC invasion, and they find that blocking the crosstalk between P2Y$_2$ and αV integrins via a peptide inhibitor blocks PDAC invasion, which may have clinical utility. Thus their study provides insights into both the basic biology of PDAC but also identifies a new target.

## Introduction

PDAC, which accounts for 90% of diagnosed pancreatic cancer cases, has the lowest survival rate of all common solid malignancies. Surgery is the only potentially curative treatment, yet more than 80% of patients present with unresectable tumors (*Kocher, 2023*). Consequently, most patients survive less

than six months after diagnosis, resulting in a five year survival rate of less than 5% when accounting for all disease stages (*Bengtsson et al., 2020*; *Kocher, 2023*). Despite continued efforts, this statistic has improved minimally in the past 50 years. Due to increasing incidence, late detection, and lack of effective therapies, pancreatic cancer is predicted to be the second most common cause of cancer-related deaths by 2040 (*Rahib et al., 2021*).

Failure to significantly improve clinical management is mainly a result of chemoresistance (*Neuzillet et al., 2017*), thus it is of vital importance to find new therapeutics that can improve patient survival. PDAC is characterized by its desmoplastic stroma, with dense fibrosis leading to impaired vascularisation and high levels of hypoxia (*Koong et al., 2000*; *Di Maggio, 2016*). Lack of oxygen leads to cellular stress and death, resulting in the release of purines such as ATP and adenosine into the tumor microenvironment (*Forrester and Williams, 1977*; *Pellegatti et al., 2008*). Extracellular ATP concentration in PDAC is 200-fold more than in normal tissue (*Hu et al., 2019*), suggesting that purinergic signaling could represent an effective therapeutic target in pancreatic cancer.

The proteins underpinning purinergic signaling comprise several highly druggable membrane proteins involved in the regulation of extracellular purines, mainly ATP and adenosine (*Burnstock and Novak, 2012*; *Boison and Yegutkin, 2019*; *Yu et al., 2021*). Extracellular ATP is known to promote inflammation (*Kurashima et al., 2012*), growth (*Ko et al., 2012*), and cell movement (*Martínez-Ramírez et al., 2016*). Contrastingly, adenosine is anti-inflammatory and promotes immunosuppression (*Schneider et al., 2021*). There are ongoing clinical trials in several cancers, including PDAC, for drugs targeting the ectonucleotidase CD73 (NCT03454451, NCT03454451) and adenosine receptor 2 A (NCT03454451) in combination with PD-1 checkpoint inhibitors and/or chemotherapy. However, a Phase II multi-cancer study evaluating an anti-CD73 and anti-PD-L1 combination was withdrawn due to minimal overall clinical activity (NCT04262388). This suggests that the oncogenic impact of purinergic signaling may act via pathways other than immunosuppression and highlights the need for the further mechanistic understanding of purinergic signaling in PDAC to exploit its full therapeutic potential.

Here, we combine bioinformatic, genetic, and drug-based approaches to identify a novel mechanism mediating ATP-driven invasion, uncovering a new therapeutic target in PDAC, a cancer of unmet clinical need. Beginning with an in-depth in silico analysis of the purinergic signaling transcriptome in PDAC, using publicly available patient and cell line databases, we build on bioinformatic data associating the purinergic receptor P2Y$_2$ with PDAC. After validating the expression of P2Y$_2$ in human PDAC, we focus on identifying the function of the receptor in cancer cells. In vitro data underline the importance of P2Y$_2$ as a strong invasive driver, using a 3D physio-mimetic model of invasion. Finally, using a super-resolution imaging technique, DNA-PAINT, we characterize the behavior of P2Y$_2$ in the membrane at the single molecule level, demonstrating the nanoscale distribution and interaction of this receptor with RGD-binding integrins in promoting pancreatic cancer invasion.

## Results

### The PDAC extracellular purinome associates with patient survival, hypoxia score and cell phenotype

The extracellular purinome encompasses 23 main surface proteins, including pannexin 1, P2X ion channels, ectonucleotidases, and the P2Y, and adenosine GPCRs (*Di Virgilio et al., 2018*; *Figure 1A*). Interrogating public databases, we determined which purinergic signaling genes significantly impact pancreatic cancer survival. First, we examined the pancreatic adenocarcinoma (PAAD) database from The Cancer Genome Atlas (TCGA; n=177 patients), analyzing overall survival hazard ratios based on purinergic signaling gene expression (*Figure 1B*). Expression of five purinergic genes correlated with decreased patient survival, with high *P2RY2* expression being associated with the highest hazard ratio (2.99, 95%, CI: 1.69–5.31, log-rank p=8.5 × 10$^{-5}$). We then examined the mutational profile and mRNA expression level of purinergic genes in patients. Using cBioPortal (*Gao, 2013*), we generated OncoPrints of purinergic signaling genes from PAAD TCGA samples (*Figure 1—figure supplement 1A*), observing few genetic alterations in 0–3% of tumors and a heterogeneous percentage of tumors with high mRNA expression (z-score >1) for each purinergic gene. PDAC molecular subtypes associated with purinergic signaling genes were varied (*Supplementary file 1*). In the Bailey model, most genes were related to the immunogenic subtype except for *NT5E*, *ADORA2B*, *PANX1,* and *P2RY2*, which are

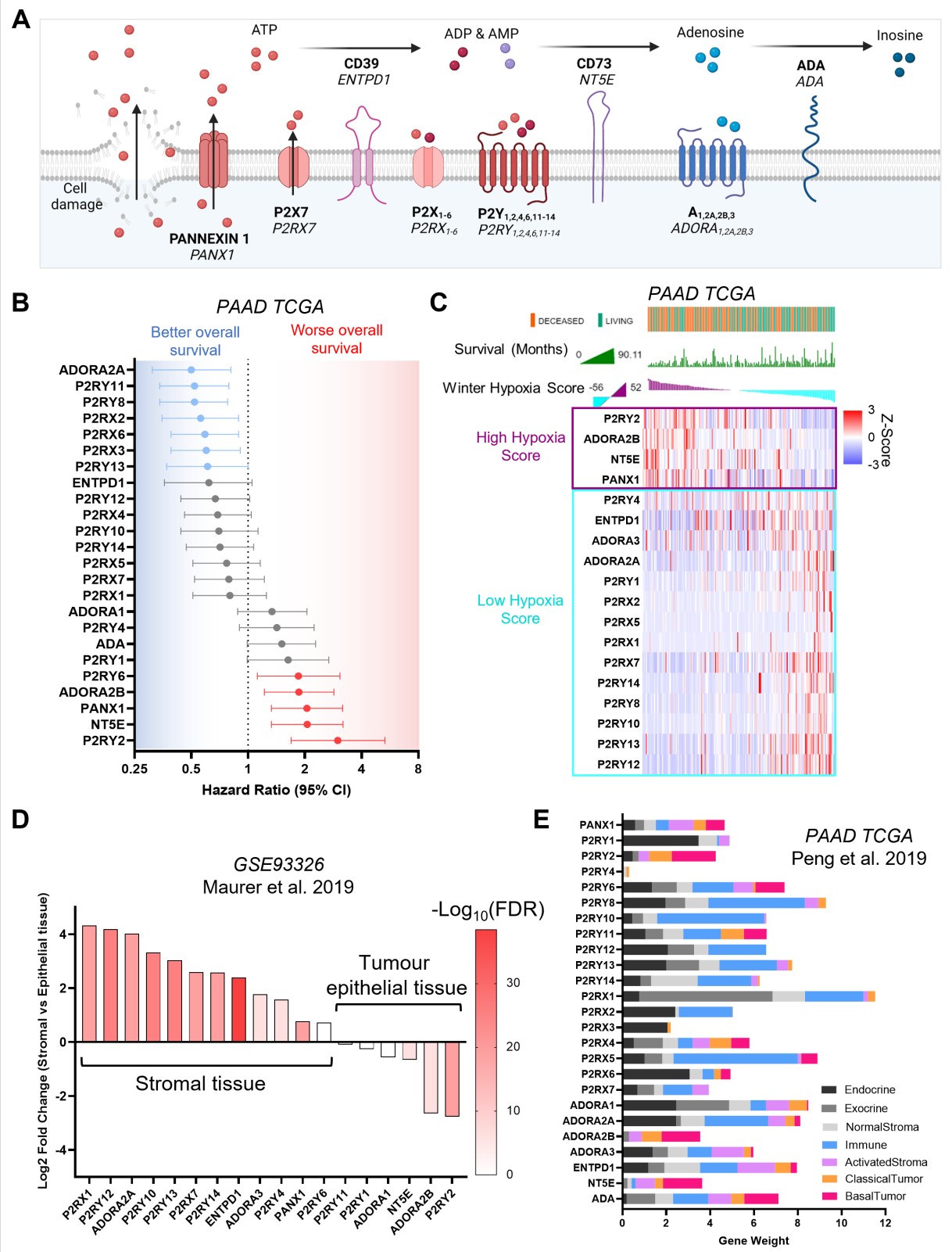

**Figure 1.** Characterization of purinergic signaling in pancreatic adenocarcinoma. (**A**) Purinergic signaling proteins and gene names. (**B**) Hazard ratios of overall survival were calculated using KMPlot and the pancreatic adenocarcinoma (PAAD) The Cancer Genome Atlas (TCGA) cohort (n=177) for different purinergic genes. Statistically significant hazard ratios (log-rank p-value) are highlighted in red for worse survival and in blue for better survival. (**C**) Heatmap of purinergic genes significantly correlated (q<0.05) to high (purple) or low (light blue) Winter hypoxia scores in the PAAD TCGA data set.

*Figure 1 continued on next page*

*Figure 1 continued*

Overall survival status and overall survival in months are shown at the top, and samples are ranked using the Winter Hypoxia score (Generated with cBioPortal). (**D**) Differential expression analysis of 60 paired stromal and tumor tissue microdissections (GSE93326) showing significantly differentially expressed purinergic genes in stromal or tumor epithelial tissue. (**E**) Gene weights for purinergic genes representing the relevance of each gene to each cell type compartment, obtained from DECODER PDAC TCGA deconvolution analysis.

The online version of this article includes the following figure supplement(s) for figure 1:

**Figure supplement 1.** Characterization of purinergic genes in pancreatic adenocarcinoma.

related to squamous (*Bailey et al., 2016*). Collisson molecular subtyping showed several purinergic genes associated mostly with quasimesenchymal and exocrine subtypes (*Collisson et al., 2011*). The Moffitt subtypes were not strongly associated with purinergic genes except for *ADA*, *NT5E*, *P2RY6*, *P2RY2,* and *PANX1* associated with the Basal subtype (*Moffitt et al., 2015*).

PDAC is known for its hypoxic environment (*Koong et al., 2000*; *Yuen and Díaz, 2014*), which is associated with worse overall survival (p=0.002, *Figure 1—figure supplement 1B*); hypoxia can lead to cellular stress and death, resulting in the increase of extracellular purines (*Forrester and Williams, 1977*). The Winter (*Winter et al., 2007*), Ragnum (*Ragnum et al., 2015*), and Buffa (*Buffa et al., 2010*) hypoxia scores were used to examine the correlation between the expression of purinergic genes and hypoxia in the PAAD TCGA database (*Figure 1—figure supplement 1C*). Samples were divided into low (n=88) or high (n=89) hypoxia score, using the median hypoxia score to perform a differential expression analysis. CD73 (*NT5E*), adenosine A2B receptor (*ADORA2B*), and P2Y$_2$ (*P2RY2*) mRNA expression associated strongly with the high hypoxia score group for all three hypoxia scores (log$_2$ ratio >0.5, FDR < 0.0001). P2Y$_2$ had the highest log$_2$ ratio in all hypoxia signatures compared to other purinergic genes. With a more extensive gene signature, the Winter hypoxia score (99 genes) allowed for a more comprehensive relative hypoxia ranking of tumor samples, compared to Ragnum (32 genes) and Buffa (52 genes) signatures. Hence, we used cBioPortal (*Gao, 2013*) to generate a transcriptomic heatmap of purinergic genes, ranked using the Winter hypoxia score and overlaid with overall survival data (*Figure 1C*). Taken together, these results show a direct correlation between Winter hypoxia score and decreased overall survival for high hypoxia score-related purinergic genes.

We hypothesized that genes related to high hypoxia scores would be expressed preferentially in the tumor cell compartment, as PDAC cells inhibit angiogenesis, causing hypo-vascularisation in the juxta-tumoral stroma (*Di Maggio, 2016*). Mining published RNA-seq data from 60 paired PDAC samples of stroma and tumor microdissections (GSE93326) (*Maurer et al., 2019*) and performing differential expression analysis, we observed that most genes related to high Winter hypoxia scores (*P2RY2*, *ADORA2B,* and *NT5E*) were expressed in the tumor epithelial tissue (*Figure 1D*), except for *PANX1*, encoding for pannexin 1, which is involved in cellular ATP release (*Bao et al., 2004*).

To elucidate the cell type-specific purinergic expression landscape, we used published data from TCGA PAAD compartment deconvolution, using DECODER (*Peng et al., 2019*) to plot purinergic gene weights for each cell type compartment (*Figure 1E*). The findings recapitulated the cell specificity data obtained from tumor microdissection analysis (*Maurer et al., 2019*; *Figure 1D*). Expression of purinergic genes in cancer cells was confirmed by plotting Z-scores of mRNA expression of PDAC cell lines from the cancer cell line encyclopedia (*Ghandi et al., 2019*) (CCLE; *Figure 1—figure supplement 1D*). Moreover, the expression of purinergic genes in normal tissue from the Genotype-Tissue Expression (GTEx) database compared to cancer tissue (PAAD TCGA) also mimicked the results found with DECODER (*Figure 1—figure supplement 1E*). *P2RY2*, encoding P2Y$_2$ - a GPCR activated by ATP and UTP, was shown to be the purinergic gene most highly associated with cancer cell-specific expression in all our independent analyses (*Figure 1D and E*; *Figure 1—figure supplement 1D, E*). *P2RY2* additionally showed the strongest correlation with all hypoxia scores (*Figure 1C*; *Figure 1—figure supplement 1C*). Most importantly, of all purinergic genes, *P2RY2* expression had the biggest adverse impact on patient survival (*Figure 1B*). These independent in silico analyses encouraged us to explore the influence of P2Y$_2$ on pancreatic cancer cell behavior.

## *P2RY2* is expressed in cancer cells and causes cytoskeletal changes

To validate our bioinformatic findings, based on microdissections from a 60 patient cohort (GSE93326) and from the deconvolution of 177 PAAD tissues from the TCGA, we performed RNAscope on human PDAC samples. This corroborated P2Y$_2$ mRNA expression as being localized to the epithelial tumor

cell compartment and not stroma, normal epithelium, or endocrine tissues (n=3, representative images of 2 different patients shown in *Figure 2A* and *Figure 2—figure supplement 1A*), matching our findings from larger publicly available cohorts, including $P2Y_2$ IHC data from 264 patients in the Renji cohort (*Hu et al., 2019*). $P2Y_2$ is known to be expressed at low levels in normal tissues but interestingly RNAscope did not detect this. This data suggests (1) the lower limits of the technique compounded by the challenge of RNA degradation in pancreatic tissue and (2) supports that in tumor tissue where it was detected there was indeed overexpression of $P2Y_2$, in line with the bioinformatic data. Interrogating single-cell $P2Y_2$ RNA expression in normal PDAC from https://www.proteinatlas.org/ (*Karlsson et al., 2021*), expression was found at low levels in several cells types, for example in endocrine cells and macrophages (*Figure 2—figure supplement 1B*). Using GEPIA (*Tang et al., 2017*), we analyzed PAAD TCGA and GTEx mRNA expression of tumor (n=179) and normal samples (n=171). Tumor samples expressed significantly higher (p<0.0001) $P2Y_2$ mRNA levels compared to the normal pancreas (*Figure 2B*). Kaplan-Meier analysis from PAAD TCGA KMplot (*Lánczky and Győrffy, 2021*) showed a significant decrease in median overall survival in patients with high $P2Y_2$ mRNA expression (median survival: 67.87 vs 17.27 months) (*Figure 2C*).

To predict $P2Y_2$ function in PDAC, we performed gene set enrichment analysis (GSEA) of high vs low mRNA expressing $P2Y_2$ tumor samples, divided by the median expression, for PAAD TCGA (n=177) and the PDAC Clinical Proteomic Tumor Analysis Consortium (CPTAC) (n=140) databases. The top gene set enriched in the PANTHER pathway database in both cohorts was the 'integrin signaling pathway' (*Figure 2D*). The top four enriched gene sets from the Gene Ontology 'Molecular function' functional database were associated with cell adhesion molecule binding, the cytoskeleton, protease binding, and extracellular matrix binding (*Figure 2—figure supplement 1C*). As preliminary validation of the GSEA results in vitro, we used the PDAC cell line AsPC-1, transduced with Lifeact, a peptide that fluorescently labels filamentous actin structures (*Riedl et al., 2008*), and monitored cell morphology using the Incucyte live-cell analysis system. Cells treated with ATP (100 μM) showed cytoskeletal rearrangements which were blocked by the selective $P2Y_2$ antagonist AR-C118925XX (AR-C; 5 μM; *Figure 2E*; *Muoboghare et al., 2019*). Exposing cells to ATP at 100 μM resulted in the biggest change in cell area when testing six concentrations from 0.01 to 1000 μM (*Figure 2—figure supplement 1D*). ATP-driven morphological changes were fully reversed at 5 X (5 μM) the $IC_{50}$ of AR-C (1 μM), while AR-C on its own had no effect on cell morphology (*Figure 2—figure supplement 1E*).

$P2Y_2$ is the only P2Y GPCR possessing an RGD motif, located in the first extracellular loop (*Figure 2F*). $P2Y_2$ has been shown to interact with αV integrins through this RGD motif (*Erb et al., 2001*), but the significance of this interaction has not been explored in cancer. Immunofluorescence (IF) showed colocalization of integrin αV and $P2Y_2$ in the PDAC cell lines AsPC-1 as well as PDAC cell lines with strong epithelial morphology, BxPC-3 and CAPAN-2, while MIA PaCa-2 cells showed low expression of both proteins, and PANC-1 showed high integrin αV and low $P2Y_2$, matching CCLE data (*Figure 2G*; *Figure 2—figure supplement 1F, G*). We hypothesized that $P2Y_2$, through its RGD motif, could engage αV integrins in cancer cells in the presence of ATP, leading to increased migration and invasion.

## Targeting $P2Y_2$ and its RGD motif decreases ATP-driven invasion in PDAC cell lines

To evaluate the impact of $P2Y_2$ in pancreatic cancer cell invasion, we used a 3D hanging drop spheroid model (*Murray et al., 2022*). PDAC cell lines were combined with stellate cells in a ratio of 1:2 (*Kadaba et al., 2013*), using an immortalized stellate cell line, PS-1 (*Froeling et al., 2009*) to form spheres (*Figure 3A*), recapitulating the ratios of the two biggest cellular components in PDAC. Stellate cells are crucial for successful hanging drop sphere formation (*Figure 3—figure supplement 1A*) and cancer cell invasion (*Murray et al., 2022*). Spheres were embedded in a Collagen type I and Matrigel mix and cultured for 48 hr until imaging and fixing (*Figure 3A*). Given that extracellular ATP concentration in tumors is in the hundred micromolar range (*Pellegatti et al., 2008*), spheres were treated with $P2Y_2$ agonists ATP and UTP (100 μM). Both nucleotides increased invasion of the PDAC cell line AsPC-1 significantly compared to vehicle control (p<0.0001 and p=0.0013, respectively), and this was blocked by the $P2Y_2$ selective antagonist AR-C (5 μM, p=0.0237, and p=0.0133; *Figure 3B and C*; *Figure 3—figure supplement 1B*). Treating spheres with AR-C on its own did not show significant effects on invasion (*Figure 3—figure supplement 1B*). Importantly, a non-hydrolyzable ATP

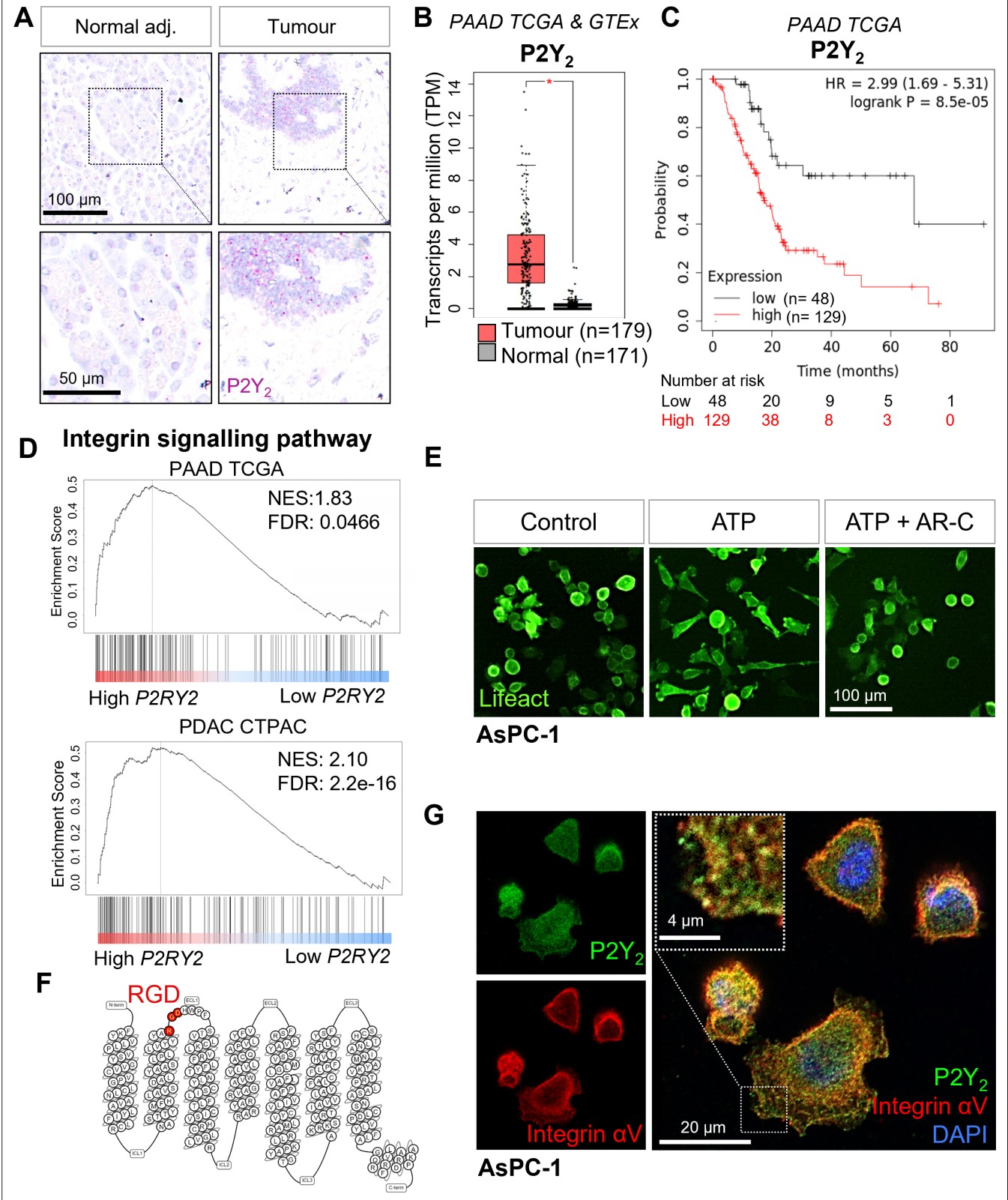

**Figure 2.** Expression of P2Y$_2$ is specific to cancer cells, correlated with decreased overall survival in patients, and drives cytoskeletal rearrangements. (**A**) RNAscope in-situ hybridization of P2Y$_2$ mRNA expression (magenta) in tumor and matching normal adjacent tissue. (**B**) P2Y$_2$ mRNA expression in tumor (TCGA) and normal (GTEx) pancreatic tissue samples (*p<0.0001). Graph generated using GEPIA. (**C**) Kaplan-Meier plot comparing patients with high vs low expression of P2Y$_2$ in the Pancreatic Adenocarcinoma (PAAD) The Cancer Genome Atlas (TCGA) cohort. Graph generated using KMplot.

*Figure 2 continued on next page*

Figure 2 continued

(**D**) Top result of a GSEA (performed with WebGestalt) of two different pancreatic adenocarcinoma patient cohorts (PAAD TCGA and PDAC CPTAC) for the PANTHER pathway functional database. (**E**) Incucyte images of the pancreatic cancer cell line AsPC-1 12 hr after treatment with 100 µM ATP alone or with 5 µM AR-C (P2Y$_2$ antagonist). Cells are transduced with Lifeact to visualize f-actin (green). (**F**) Schematic of the amino acid sequence of P2Y$_2$ showing an RGD motif in the first extracellular loop (image generated in http://gpcrdb.org/). (**G**) IF staining of P2Y$_2$ (green), integrin αV (red), and DAPI (blue) in AsPC-1 cells showing colocalization of P2Y$_2$ and integrin αV (yellow).

The online version of this article includes the following figure supplement(s) for figure 2:

**Figure supplement 1.** mRNA and protein expression of P2Y$_2$ in pancreatic ductal adenocarcinoma (PDAC) cells.

(ATPγS;100 µM) showed similar effects to ATP, implicating ATP and not its metabolites as the cause of the invasion (*Figure 3—figure supplement 1C*). Of note, IF staining of PS-1 cells showed negligible expression of P2Y$_2$ (*Figure 3—figure supplement 1D*). To determine whether integrin association was necessary for ATP-driven invasion, we treated spheres with 10 µM cyclic RGDfV peptide (cRGDfV), which binds predominantly to αVβ3 to block integrin binding to RGD motifs (*Kapp et al., 2017*), such as that in P2Y$_2$ (*Ibuka et al., 2015*). cRGDfV treatment reduced ATP-driven motility significantly, both in 3D spheroid invasion assays (p<0.0001) (*Figure 3B and C*) and in 2D Incucyte migration assays (*Figure 3—figure supplement 1E, F*) as did treatment with AR-C. To ensure that this behavior was not restricted to AsPC-1 cells, experiments were corroborated in the epithelial-like BxPC-3 cell line (*Figure 3—figure supplement 1G, H*; *Tan et al., 1986*).

To further verify that ATP-driven invasion was dependent on P2Y$_2$, we silenced P2Y$_2$ expression in AsPC-1 cells using siRNA (*Figure 3D*; *Figure 3—figure supplement 1I*), abrogating the invasive response to ATP (p<0.0001). P2Y$_2$ involvement in this phenomenon was confirmed by generating a P2Y$_2$ CRISPR-Cas9 AsPC-1 cell line (P2Y$_2$$^{CRISPR}$), which displayed a significant decrease in invasion compared to a control guide RNA CRISPR cell line (CTR$^{CRISPR}$) in both ATP-treated (p<0.0001) and non-treated (p=0.0005) conditions (*Figure 3F and E*). Additionally, we tested the off-target effects of AR-C in AsPC-1 P2Y$_2$$^{CRISPR}$ spheres and confirmed no significant difference in invasion compared to the control (*Figure 3—figure supplement 1J*). Together, these findings demonstrate that P2Y$_2$ is essential for ATP-driven cancer cell invasion.

To determine the importance of the RGD motif of P2Y$_2$ in ATP-driven invasion, we obtained a mutant P2Y$_2$ construct, where the RGD motif was replaced by RGE (P2Y$_2$$^{RGE}$), which has less affinity for αV integrins (*Erb et al., 2001*). This mutant was transfected into AsPC-1 P2Y$_2$$^{CRISPR}$ cells and compared to cells transfected with wild-type P2Y$_2$ (P2Y$_2$$^{RGD}$; *Figure 3—figure supplement 1K*). Only spheres containing P2Y$_2$$^{RGD}$ transfected cells demonstrated a rescue of the ATP-driven invasive phenotype (p<0.0001; *Figure 3G and H*), with P2Y$_2$$^{RGE}$ spheres not responding to ATP treatment. To ensure this behavior was not influenced by off-target CRISPR effects, we repeated the experiment in PANC-1 cell line, which expresses very low levels of P2Y$_2$, but high levels of integrin αV (*Figure 2—figure supplement 1F*, G). No ATP-driven invasion was observed in PANC-1 cells transfected with an empty vector (EV) or with P2Y$_2$$^{RGE}$ (*Figure 3I and J*). Only when transfecting PANC-1 cells with P2Y$_2$$^{RGD}$ was ATP-driven invasion observed (p<0.0001). These results demonstrate that the RGD motif of P2Y$_2$ is required for ATP-driven cancer cell invasion.

## DNA-PAINT reveals RGD-dependent changes in P2Y$_2$ and integrin αV surface expression

To interrogate how P2Y$_2$ interacts with αV integrins, we examined the nanoscale organization of P2Y$_2$ and αV proteins under different treatment conditions using a multi-color quantitative super-resolution fluorescence imaging method, DNA-PAINT. DNA-PAINT is a single-molecule localization microscopy (SMLM) method based on the transient binding between two short single-stranded DNAs - the 'imager' and 'docking' strands. The imager strand is fluorescently labeled and freely diffusing in solution, whilst the docking strand is chemically coupled to antibodies targeting the protein of interest. For DNA-PAINT imaging of P2Y$_2$ and integrin αV, proteins were labeled with primary antibodies chemically coupled to orthogonal docking sequences featuring a repetitive (ACC)n or (TCC)n motif, respectively (*Figure 4A*). The benefit of such sequences is to increase the frequency of binding events, which in turn allows the use of relatively low imager strand concentrations without compromising overall imaging times, whilst achieving a high signal-to-noise ratio and single-molecule localization precision (*Strauss and Jungmann, 2020*).

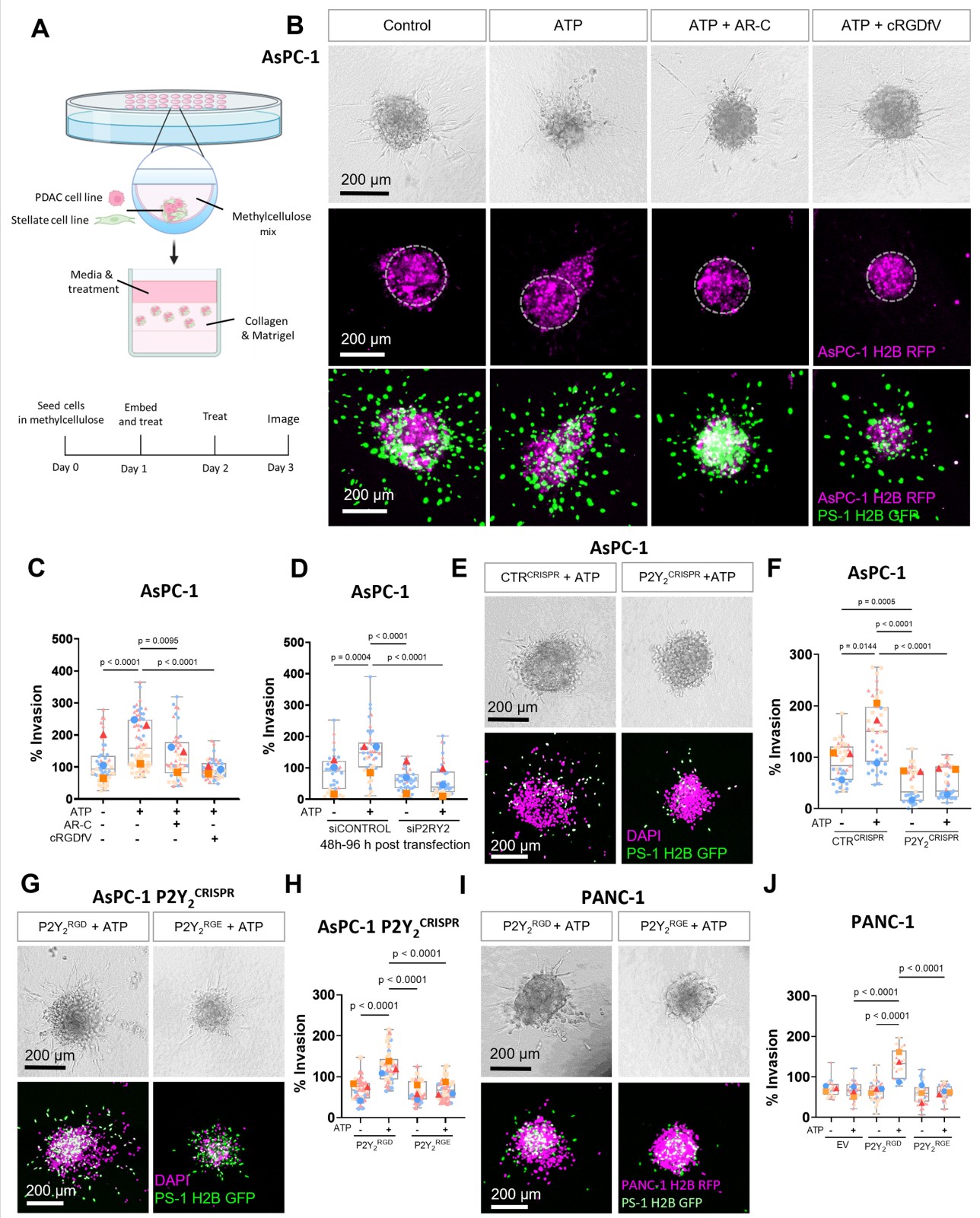

**Figure 3.** The RGD motif in P2Y$_2$ is required for extracellular ATP-driven cancer cell invasion. (**A**) Schematic diagram of the hanging drop sphere model for 3D sphere invasion assays. (**B**) Bright field and fluorescent images of spheres formed using AsPC-1 cells (magenta) with a histone 2B (H2B) tagged with a red fluorescent protein (RFP) and the stellate cell line PS-1 (green) with H2B tagged with a green fluorescent protein (GFP). Middle panel shows AsPC-1 cells in spheres with a dotted line highlighting the central sphere area. Spheres were treated with vehicle control or 100 μM ATP alone or with

*Figure 3 continued on next page*

*Figure 3 continued*

5 µM AR-C or 10 µM cRGDfV. The quantification is shown in (**C**) using SuperPlots, where each color represents a biological repeat (n=3) and the larger points represent the mean % Invasion for each repeat. (**D**) Quantification of spheres formed by AsPC-1 cells transfected with a control siRNA or P2Y$_2$ siRNA and treated with or without 100 µM ATP. (**E**) Bright field and fluorescent images of spheres formed by AsPC-1 cells subjected to CRISPR/Cas9 gene disruption using a control guide RNA (CTR$^{CRISPR}$) or P2Y$_2$ guide RNAs (P2Y$_2$$^{CRISPR}$) and treated with or without 100 µM ATP. Quantification in (**F**). (**G, I**) Bright field and fluorescent images of AsPC-1 P2Y$_2$$^{CRISPR}$ cells or PANC-1 cells (respectively) transfected with wild-type *P2RY2* (P2Y$_2$$^{RGD}$) or mutant *P2RY2$^{D97E}$* (P2Y$_2$$^{RGE}$) treated with or without 100 µM ATP and its quantification in (**H**) and (**J**), respectively. Statistical analysis with Kuskal-Wallis multiple comparison tests.

The online version of this article includes the following source data and figure supplement(s) for figure 3:

**Figure supplement 1.** Invasion and migration experiments in Pancreatic ductal adenocarcinoma (PDAC) cell lines.

**Figure supplement 1—source data 1.** Labeled uncropped blot of *Figure 3—figure supplement 1I*.

**Figure supplement 1—source data 2.** Full unedited blot of *Figure 3—figure supplement 1I*.

The repetitive binding of imager and docking DNA strands in DNA-PAINT causes the same protein to be detected multiple times with nearly identical coordinates, originating a cluster of single-molecule localization around the true position of the protein. In contrast to other SMLM approaches, it is possible to take advantage of the DNA-binding kinetics to stoichiometrically calculate the number of proteins detected in each cluster of single molecule localizations, via an approach known as qPAINT (*Schnitzbauer et al., 2017*). As exemplified in *Figure 4B* (and detailed in the methods section), qPAINT relies on the first-order binding kinetics between the individual imager and docking strands to determine the number of copies of a protein that reside within a cluster of single-molecule localizations. The qPAINT index histograms obtained from P2Y$_2$ and αV DNA-PAINT data sets were fitted with a multi-peak Gaussian function, identifying peaks located at multiples of a qPAINT index value of $q_{i,1}$ 0.011 Hz and 0.009 for the P2Y$_2$ and αV docking-imager pairs, respectively (*Figure 4C*). These values were thus used to quantify the exact number of P2Y$_2$ and αV proteins in all the clusters of single-molecule localization in the DNA-PAINT data sets. By combining qPAINT with spatial statistics, we recovered a good estimation of the ground truth position of all the proteins in the DNA-PAINT data and quantified protein clustering.

We have previously analyzed GPCR oligomerization quantitatively using DNA-PAINT super-resolution microscopy of P2Y$_2$ in AsPC-1 cells (*Joseph et al., 2021*), where we observed a decrease in P2Y$_2$ oligomerization upon AR-C treatment. Hence, we questioned whether the RGD motif in P2Y$_2$ affected receptor distribution and clustering. We imaged AsPC-1 P2Y$_2$$^{CRISPR}$ cells transfected with P2Y$_2$$^{RGD}$ or P2Y$_2$$^{RGE}$ in the absence or presence of 100 µM ATP for 1 hr (*Figure 4D*), observing a 42% decrease in the median density of P2Y$_2$ proteins at the membrane when P2Y$_2$$^{RGD}$ cells were treated with ATP, compared to control (p<0.0001; *Figure 4E*). In contrast, although a slight decrease in the density of P2Y$_2$ proteins on P2Y$_2$$^{RGE}$ cells was observed following ATP treatment, this was not statistically significant (p=0.1570). The density of P2Y$_2$ proteins and protein clusters in both P2Y$_2$$^{RGD}$ and P2Y$_2$$^{RGE}$ controls were equivalent (*Figure 4E and F*; p>0.9999), indicating similar expression of the receptor at the surface in both control conditions. Interestingly, the density of P2Y$_2$ clusters decreased significantly in both conditions when treated with ATP (*Figure 4F*; 43% decrease, p<0.0001 for P2Y$_2$$^{RGD}$, and 48% decrease, p=0.0002 for P2Y$_2$$^{RGE}$). We repeated these studies with normal AsPC-1 cells (untransfected and with unaltered P2Y$_2$ expression) treated with ATP +/-cRGDfV, only observing a reduction of P2Y$_2$ at the membrane with ATP alone (68% decrease, p<0.0001), while co-treatment with cRGDfV prevented this change (p>0.9999; *Figure 4—figure supplement 1A, B*). These findings highlight that the RGD motif is required for αV integrin to control P2Y$_2$ levels at the membrane.

Turning to αV integrins, we observed an increase in the density of αV molecules and αV clusters at the membrane when stimulating P2Y$_2$$^{RGD}$ with ATP (165 αV molecules/ROI, IQR = 162.75; 6.5 αV clusters/ROI, IQR = 8.75) compared to P2Y$_2$$^{RGD}$ without stimulation (58 αV molecules/ROI, IQR = 41; 2.5 αV clusters/ROI, IQR = 2; p=0.0003; *Figure 4G and H*). This phenomenon was also observed with normal AsPC-1 cells, with significantly more αV molecules and clusters (p=0.0382 and p=0.0349) detected following ATP stimulation (*Figure 4—figure supplement 1C, D*). In absence of stimulation, P2Y$_2$$^{RGE}$ transfected cells exhibited more αV molecules and clusters at the membrane (182 αV molecules/ROI, IQR = 262.75; 9 αV clusters/ROI IQR = 14) compared to P2Y$_2$$^{RGD}$ cells (p=0.0003, p=0.0024, respectively). However, treating P2Y$_2$$^{RGE}$ cells with ATP did not result in significant changes in αV molecules and clusters (p=0.7086; p=0.1846). When the number of clusters was normalized with the

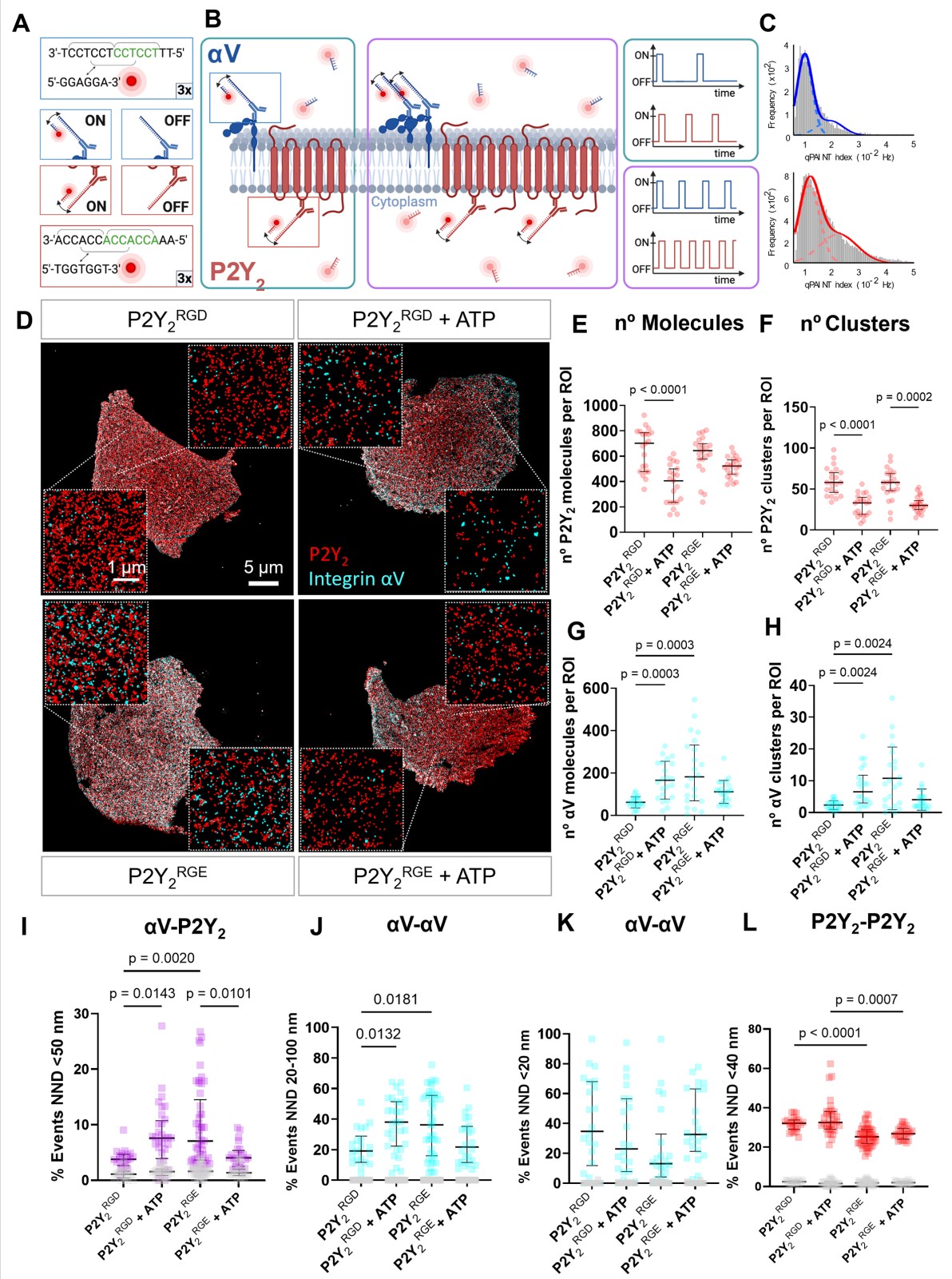

**Figure 4.** DNA-PAINT super-resolution microscopy reveals ATP and RGD-dependent changes in the number and distribution of integrin αV and P2Y₂ molecules in the plasma membrane. (**A, B**) Overview of the DNA-PAINT microscopy technique and qPAINT analysis pipeline. (**C**) Histogram of qPAINT indices for αV (blue) and P2Y₂ (red) single-molecule localization clusters. Solid lines represent multi-peak Gaussian fit. (**D**) Rendered DNA-PAINT images of AsPC-1 P2Y₂^CRISPR cells transfected with P2Y₂^RGD or P2Y₂^RGE with or without 100 µM of ATP and close-ups showing the protein maps reconstructed from

*Figure 4 continued on next page*

Figure 4 continued

DNA-PAINT localization maps of P2Y$_2$ (red) and integrin αV (cyan). The quantification of the number of proteins or protein clusters (>3 proteins) in each region of interest (ROI) are for P2Y$_2$ (red) (**E**) and (**F**), respectively and integrin αV (cyan) (**G**) and (**H**), respectively. Quantification of protein proximity using the nearest neighbor distance (NND), with the percentages of integrin αV and P2Y$_2$ proteins being <50 nm apart (**I**), between different αV integrins being 20–100 nm (**J**) or <20 nm (**K**) apart; and P2Y$_2$ from other P2Y$_2$ proteins being <40 nm apart (**H**). Statistical analysis with Kuskal-Wallis multiple comparison test of 21 4x4 µm ROIs from a minimum of 5 cell regions per condition.

The online version of this article includes the following figure supplement(s) for figure 4:

**Figure supplement 1.** Quantification of P2Y2 and integrin αV at the membrane using DNA-PAINT.

**Figure supplement 2.** Schematic diagram of nearest neighbor distance (NND) distances and NND histograms.

number of αV molecules, to obtain the percentage of αV in clusters (*Figure 4—figure supplement 1E*), there was no significant difference between conditions (p>0.9999), indicating that the increase in the number of αV clusters was due to an increase in the number of αV proteins at the membrane. Of note, the percentage of P2Y$_2$ clusters significantly decreased in P2Y$_2$$^{RGE}$ cells when treated with ATP compared to all other conditions (*Figure 4—figure supplement 1F*). Taken together, these data indicate an RGD motif-dependent function of activated P2Y$_2$ in localizing integrin αV to the membrane.

Nearest neighbor distance (NND) was used to analyze homo and heterotypic protein-protein interactions between P2Y$_2$ and αV. NND ranges were selected by using the approximate dimension of the antibodies (~14 nm) (*Tan, 2008*), integrins (5–10 nm) (*Lepzelter et al., 2012*), and GPCRs (~3 nm) (*Figure 4—figure supplement 2A*) and corroborating them with the NND histograms (*Figure 4—figure supplement 2B*) to predict the NND range in nm indicating a protein-protein interaction. We detected a higher percentage of integrin αV proteins in <50 nm proximity to P2Y$_2$ in P2Y$_2$$^{RGD}$ cells following ATP stimulation (*Figure 4I*; 103% increase, p=0.0143). In contrast, P2Y$_2$$^{RGE}$ cells stimulated with ATP showed a 43% decrease (p=0.0101) in αV molecules in close proximity to P2Y$_2$ in comparison to unstimulated cells. Analyzing the percentage of αV proteins with NND in the 20–100 nm range, we saw a similar pattern (*Figure 4J*). ATP-stimulated P2Y$_2$$^{RGD}$ and unstimulated P2Y$_2$$^{RGE}$ cells showed an increased percentage of αV proteins spaced at this range compared to untreated P2Y$_2$$^{RGD}$ cells (98% increase with p=0.0132 and 89% increase with p=0.0181). No significant changes were observed in NND of <20 nm between αV proteins in any of the conditions (*Figure 4K*). In contrast, P2Y$_2$$^{RGD}$ molecules were in significantly closer proximity to each other compared to P2Y$_2$$^{RGE}$ in control and stimulated conditions (p<0.0001 and p=0.007) (*Figure 4L*). In summary, our SMLM studies demonstrate a reciprocal interaction between αV integrin and P2Y$_2$ receptors, where P2Y$_2$ can alter integrin localization to the plasma membrane while αV integrins influence activated P2Y$_2$ membrane localization.

## The RGD motif in P2Y$_2$ is involved in integrin signaling

There is growing evidence of the importance of endosomal GPCR signaling and its potential relevance in disease and therapeutic opportunities (*Calebiro and Godbole, 2018*). As we identified the RGD motif in P2Y$_2$ having a possible role in receptor internalization, integrin dynamics, and invasion, we proceeded to look at integrin signaling through phosphorylation of FAK (p-FAK) and ERK (p-ERK) from 0 to 1 hr after treating with 100 µM ATP. AsPC-1 cells displayed a significant increase of FAK and ERK phosphorylation after 15 min of ATP stimulation, which was abrogated by concomitant targeting of P2Y$_2$ with AR-C (*Figure 5A*). When impairing the RGD motif function in P2Y$_2$ with cRGDfV or by transfecting AsPC-1 P2Y$_2$$^{CRISPR}$ cells with the P2Y$_2$$^{RGE}$ mutant, p-FAK, and p-ERK levels decreased (*Figure 5B and C*). Collectively, targeting the RGD motif in P2Y$_2$ impairs receptor signaling and inhibits pancreatic cancer cell invasion.

## Discussion

Improved molecular understanding of PDAC is vital to identify effective therapeutic approaches to improve patient survival. Purinergic signaling includes many druggable targets that have been related to hypoxia (*Synnestvedt et al., 2002*), immunosuppression (*Fong et al., 2020*), and invasion (*Li et al., 2015*), but have been relatively underexplored in PDAC. In this study, we used publicly available databases to identify purinergic signaling genes that could be promising targets for PDAC, determining P2Y$_2$ as a driver of pancreatic cancer cell invasion. Extracellular ATP stimulated invasion in a 3D spheroid model of PDAC; an effect blocked by targeting P2Y$_2$ genetically and pharmacologically.

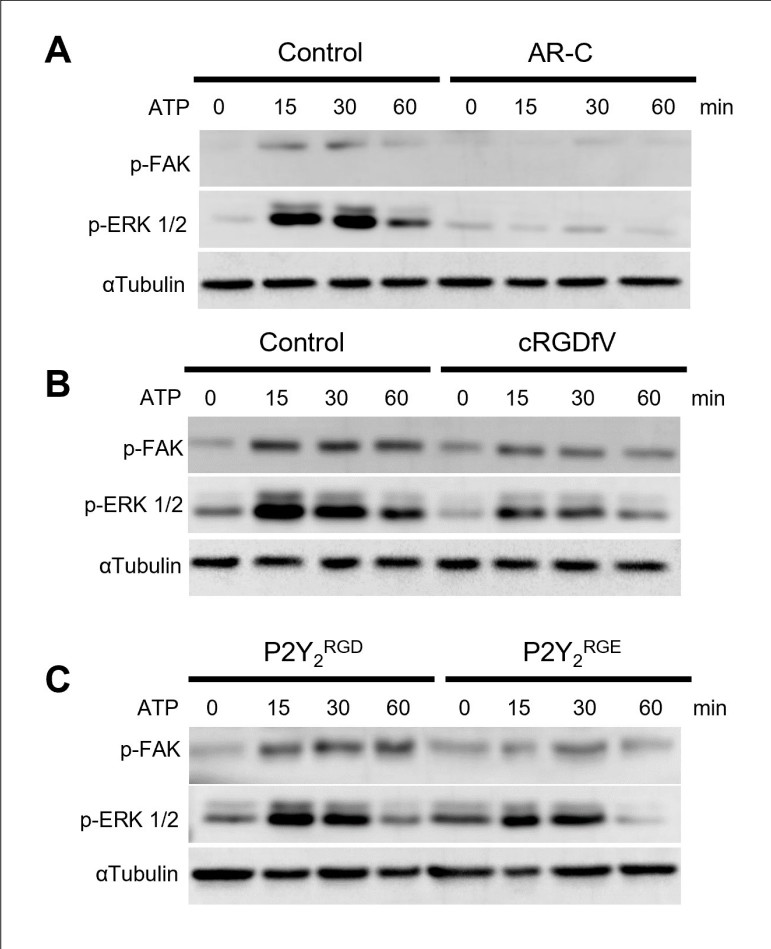

**Figure 5.** The RGD motif in P2Y$_2$ is involved in FAK/ERK signaling. (**A, B**) Western blots of phosphorylated FAK (p-FAK) and ERK (p-ERK) of AsPC-1 cells treated with ATP or pre-treated for 30 min with AR-C (5 µM) or cRGDfV (10 µM), respectively and treated with ATP for 60 min. (**C**) Western blot of AsPC-1 P2Y$_2^{CRISPR}$ cells transfected with P2Y$_2^{RGD}$ or P2Y$_2^{RGE}$ and treated with ATP for 60 min. Representative images of three biological replicates.

The online version of this article includes the following source data for figure 5:

**Source data 1.** Labeled uncropped blots of *Figure 5*.

**Source data 2.** Full unedited blots of *Figure 5*.

Mechanistically, we identified that the RGD motif in the first extracellular loop of P2Y$_2$ is required for ATP-driven cancer invasion. Importantly, quantitative DNA-PAINT super-resolution fluorescence microscopy revealed the role of this RGD motif in orchestrating the number of P2Y$_2$ and αV integrin proteins at the plasma membrane, upon ATP stimulation.

Purinergic signaling has been associated classically with hypoxia and immune function in cancer (*Di Virgilio et al., 2018*). One of the first reports of hypoxia-inducing ATP release in cells identified an increase of extracellular ATP in rat heart cells when kept in hypoxic conditions (*Forrester and Williams, 1977*). PDAC is a highly hypoxic cancer, with high levels of ATP reported in the tumor interstitial fluid of human and mouse PDAC tissues compared to healthy tissues (*Hu et al., 2019*). This vast release of ATP results in immune-mediated inflammatory responses via immune cells expressing purinergic signaling receptors (*Chiarella et al., 2021*). Expression of most purinergic genes was associated predominantly with immune cells, immunogenic PDAC subtype, and low hypoxia scores (*Figure 1C and E*). In contrast, expression of genes correlated with worse survival and hypoxia (*PANX1, NT5E, ADORA2B,* and *P2RY2*) was associated with tumor cells and the squamous PDAC subtype, correlating with hypoxia, inflammation, and worse prognosis (*Bailey et al., 2016*). The role of CD73 in PDAC has been examined in several studies (*Yu et al., 2021*) (NCT03454451, NCT03454451). In contrast, the

adenosine $A_{2B}$ receptor has not been well studied. Adenosine $A_{2B}$ receptor requires larger agonist concentrations for activation compared to other receptors in the same family, such as adenosine $A_{2A}$ (***Bruns et al., 1986***; ***Xing et al., 2016***), and receptor expression has been reported to increase when cells are subjected to hypoxia (***Feoktistov et al., 2004***). Moreover, HIF-1α has been shown to upregulate $A_{2B}$ and P2Y$_2$ expression in liver cancer (***Tak et al., 2016***; ***Kwon et al., 2019***). From our analyses, P2Y$_2$ was associated with the worst patient overall survival, highest patient hypoxia scores, and strongest correlation with cancer cell expression compared to other purinergic genes. These observations were supported by published immunohistochemical staining of 264 human PDAC samples, showing that P2Y$_2$ localized predominantly in cancer cells in human PDAC and that P2Y$_2$ activation with ATP led to elevated HIF-1α expression (***Hu et al., 2019***). Hence, we decided here to explore P2Y$_2$ in greater depth.

P2Y$_2$ has been associated with cancer cell growth and glycolysis in PDAC (***Ko et al., 2012***; ***Hu et al., 2019***; ***Wang et al., 2020***). Combination treatment of subcutaneous xenografts of AsPC-1 or BxPC-3 cells with the P2Y$_2$ antagonist AR-C together with gemcitabine significantly decreased tumor weight and resulted in increased survival compared to placebo or gemcitabine monotherapy control (***Hu et al., 2019***). Surprisingly, GSEA results of two different cohorts suggested a possible additional function of P2Y$_2$ in invasion. Increased glycolysis and cytoskeletal rearrangements have been linked (***Park et al., 2020***), and both events could occur downstream of P2Y$_2$ activation. P2Y$_2$ has been implicated in invasive phenotypes in prostate, breast, and ovarian cancer (***Jin et al., 2014***; ***Li et al., 2015***; ***Martínez-Ramírez et al., 2016***). Moreover, high P2Y$_2$ expression in patients was related to integrin signaling. The RGD motif in the first extracellular loop of P2Y$_2$ results in a direct interaction of P2Y$_2$ with RGD-binding integrins, particularly integrins αVβ3 and αVβ5 (***Erb et al., 2001***; ***Ibuka et al., 2015***). This interaction can exert phenotypic effects – for example, the binding of P2Y$_2$ to integrins via its RGD motif is necessary for tubule formation in epithelial intestinal cell line 3D models (***Ibuka et al., 2015***). We focus here on the importance of the RGD motif of P2Y$_2$ and its key for integrin interaction in a cancer context. We were able to abrogate ATP-driven invasion using either the P2Y$_2$ selective antagonist AR-C or by blocking P2Y$_2$-integrin complexes using the selective αVβ3 cyclic RGD-mimetic peptide inhibitor cRGDfV. Likewise, spheres made using ASPC-1 P2Y$_2$$^{CRISPR}$ or PANC-1 cells transfected with mutant P2Y$_2$$^{RGE}$, which decreases the affinity of P2Y$_2$ for integrins, did not invade in response to ATP stimulation. Altogether, these results (1) support P2Y$_2$ involvement in PDAC cell invasion, (2) show the RGD motif is essential for this function, and (3) identify the mechanism for this to be caused by P2Y$_2$-integrin complexes. Despite efforts, there are currently no clinically efficacious P2Y$_2$ antagonists, with poor oral bioavailability and low selectivity being major issues (***Neumann et al., 2022***). Our findings demonstrate that P2Y$_2$ can also be targeted by blocking its interaction with RGD-binding integrins, due to its dependence on integrins for its pro-invasive function.

GPCR-integrin crosstalk is involved in many biological processes (***Wang et al., 2005***; ***Teoh et al., 2012***). Only one study has directly examined the spatial distribution of integrins and GPCRs, however, this relied on IF analysis (***Erb et al., 2001***), where only changes in the micron scale will be perceived, hence losing information on the nanoscale distances and individual protein interactions. Here, we present a method to image integrin and GPCR dynamics using quantitative DNA-PAINT super-resolution fluorescence microscopy (***Schnitzbauer et al., 2017***), allowing spatial and quantitative assessment of P2Y$_2$ and integrin αV interactions at the single protein level. Following ATP stimulation, the number of P2Y$_2$ proteins at the plasma membrane decreased significantly after 1 hr, implying receptor internalization, in line with previous work showing P2Y$_2$ at the cell surface was reduced significantly after 1 hr of UTP stimulation (***Tulapurkar et al., 2005***). Of note, cytoskeletal rearrangements, which we have also observed upon ATP stimulation (***Figure 2E***), were required for P2Y$_2$ clathrin-mediated internalization, and authors noted that P2Y$_2$ was most likely in a complex with integrins and extracellular matrix-binding proteins. Cells expressing RGE mutant P2Y$_2$ or treated with cRGDfV, did not show significant changes in P2Y$_2$ levels at the membrane upon ATP treatment, thus implicating the RGD motif in P2Y$_2$ in agonist-dependent receptor internalization, though we have focused on motility phenotype in this work.

P2Y$_2$ affecting cell surface redistribution of αV integrin has been reported, with αV integrin clusters observed after 5 min stimulation with UTP (***Chorna et al., 2007***). We observed an increased number of αV integrin molecules and clusters 1 hr after ATP stimulation, although this increase in clusters was mainly due to the increase in the total number of αV integrins at the membrane. The distance between

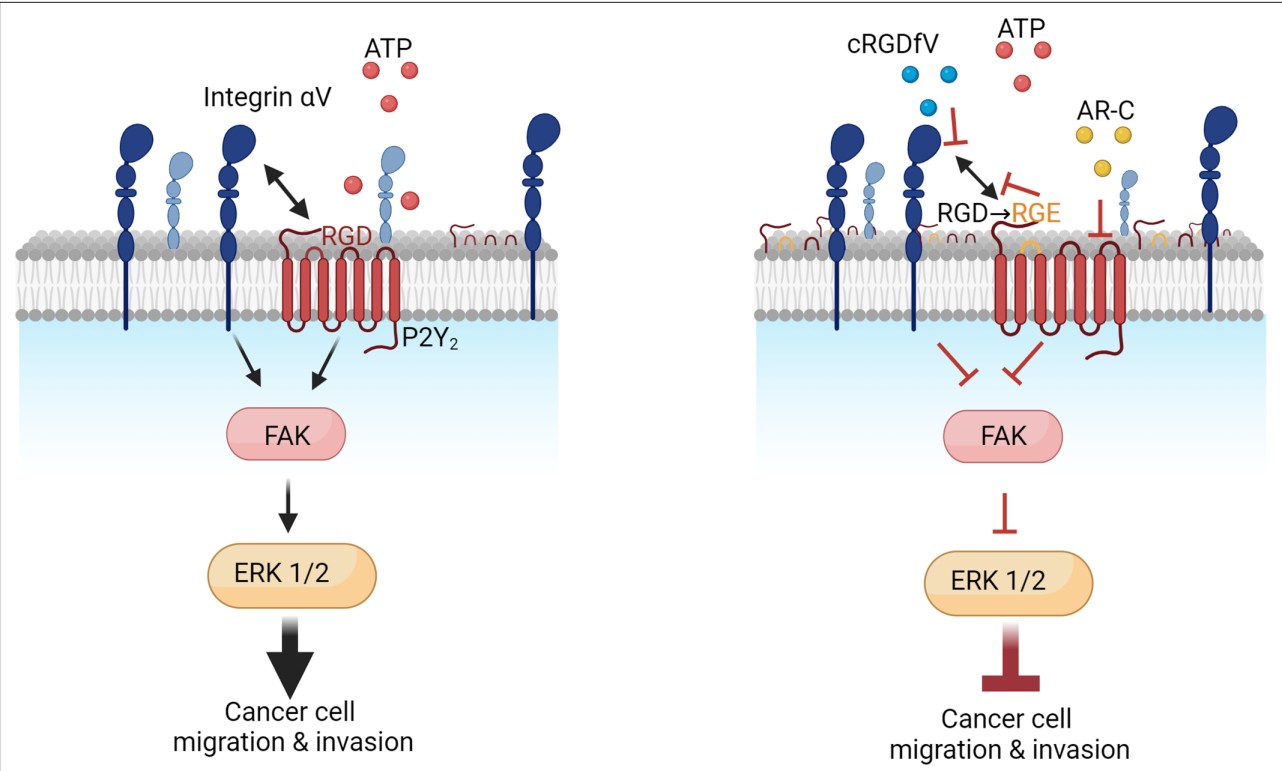

**Figure 6.** Proposed mechanism of P2Y$_2$ and integrin interactions in pancreatic cancer invasion.

αV integrin and P2Y$_2$ molecules decreased (NND <50 nm) with ATP stimulation, indicating possible interaction. In contrast, with mutant P2Y$_2$$^{RGE}$, no significant ATP-dependent changes in the number of P2Y$_2$ or αV integrin proteins at the membrane were observed. The same phenomenon was observed when treating normal AsPC-1 cells (untransfected and with no alteration to P2Y$_2$) with cRGDfV and ATP. We speculate that by reducing the ability of integrins to bind to the RGD of P2Y$_2$, through receptor internalization, RGE mutation or through cRGDfV treatment, there is less RGD-triggered integrin endocytosis, hence less integrin recycling and an increase of integrins at the cell surface. Western blot results supported our postulated role of the RGD motif in P2Y$_2$ regulating downstream integrin signaling through FAK and ERK, leading to cancer cell migration and invasion (*Figures 5 and 6*). This is the first single-molecule super-resolution study to explore integrin and GPCR dynamics and to demonstrate a requirement for integrin-P2Y$_2$ interactions in cancer cell invasion.

In summary, our study demonstrates that P2Y$_2$, via its RGD motif, has a pivotal role in ATP-induced PDAC invasion by interacting with and regulating the number of αV integrins at the plasma membrane, revealing this critical axis as a promising therapeutic target.

## Methods

### Data mining and bioinformatic analysis

Hazard ratios and the P2Y$_2$ Kaplan-Meier plot for overall survival were obtained using Kaplan-Meier Plotter (RRID:SCR_018753) (*Lánczky and Győrffy, 2021*) and the pancreatic adenocarcinoma dataset from the cancer genome atlas (PAAD TCGA, RRID:SCR_003193).

Using cBioPortal (RRID:SCR_014555) (*Gao, 2013*) and the database PAAD TCGA, mRNA differential expression analysis was performed for each Hypoxia Score (*Winter et al., 2007*; *Buffa et al., 2010*; *Ragnum et al., 2015*) by separating patients using the median hypoxia score. Results from purinergic genes were plotted in a volcano plot using VolcaNoseR (*Goedhart and Luijsterburg, 2020*). Significant hits were plotted in a heat map using cBioPortal (*Gao, 2013*). RNAseq raw counts from stromal and epithelial PDAC tissue from microdissections were downloaded from the GEO database

(GSE93326) (*Maurer et al., 2019*) and a differential expression analysis was performed using DESeq2 (RRID:SCR_015687) (*Love et al., 2014*; *Varet et al., 2016*) in R.

Gene weight results from DECODER from PDAC tissues in the TCGA database were obtained from published results (*Peng et al., 2019*). Using GEPIA (RRID:SCR_018294) (*Tang et al., 2017*), mRNA expression of purinergic genes in normal tissue from the Genotype-Tissue Expression (GTEx, RRID:SCR_013042) compared to cancer tissue (PAAD TCGA) was obtained. PDAC cell line mRNA z-scores or mRNA reads per kilobase million (RPKM) were obtained using cBioPortal and the Cancer Cell Line Encyclopaedia (CCLE, RRID:SCR_013836) data (*Gao, 2013*).

For gene set enrichment analysis (GSEA), cBioPortal was used to separate PAAD TCGA or PDAC CPTAC patients into high and low *P2RY2* by *P2RY2* median expression and perform the differential expression analysis. Log ratio values were inserted in the WEB-based Gene SeT AnaLysis Toolkit (WebGestalt, RRID:SCR_006786) (*Liao et al., 2019*), where 'GO: Molecular Function' or 'Panther' with default analysis parameters were selected.

## RNAscope in-situ hybridization

Formalin-fixed paraffin-mbedded (FFPE) sections (n=3) of PDAC with stroma and normal adjacent tissue were obtained from the Barts Pancreas Tissue Bank (Project 2021/02/QM/RG/E/FFPE). Sections were stained using the human *P2RY2* probe (853761, ACD) and the RNAscope 2.5 HD Assay-RED (ACD) following the manufacturer's instructions. Slides were imaged by NanoZoomer S210 slide scanner (Hamamatsu).

## Cell lines and cell culture

The pancreatic cancer cell lines AsPC-1 (RRID:CVCL_0152), BxPC-3 (RRID:CVCL_0186), MIA PaCa-2 (RRID:CVCL_0428) and PANC-1 (RRID:CVCL_0480), in addition to the immortalized stellate cell line PS-1 (*Froeling et al., 2009*) were kindly donated by Prof. Hemant Kocher (Queen Mary University of London). Cell lines stably expressing fluorescently labeled histone subunits (H2B) or Lifeact (*Riedl et al., 2008*) were transduced with viral supernatant obtained from HEK293T cells co-transfected with pCMVR8.2 (Addgene #12263) and pMD2.G (Addgene #12259) packaging plasmids, and either H2B-GFP (Addgene #11680), H2B-RFP (Addgene #26001), or Lifeact-EGFP (Addgene # 84383) plasmids using FuGENE transfection reagent (Promega), following manufacturer's guidelines. Successfully transduced cells were isolated using a BD FACS Aria Fusion cell sorter. AsPC-1 $P2Y_2^{CRISPR}$ cells were generated by transfecting cells with a dual gRNA (TGAAGGGCCAGTGGTCGCCGCGG and CATCAGCGTGCACCGGTGTCTGG)CRISPR-CAS9 plasmid (VectorBuilder) with an mCherry marker which was used to select successfully transfected cells as above. Clonal expansion of single sorted cells was achieved with serial dilution cloning. Clones were evaluated by IF for $P2Y_2$ compared to parental AsPC-1 cells. Cell lines were grown at 37 °C with 5% $CO_2$ in DMEM (Gibco), RPMI-1640 (Gibco), or DMEM/F-12 (Sigma) supplemented with 10% fetal bovine serum (Sigma). Cells were monitored for mycoplasma contamination every six months.

## Cell fixation and immunofluorescent staining

Cells were seeded on coverslips placed in a six well-plate (Corning) and fixed the next day in 4% paraformaldehyde (LifeTech) for 30 min and washed 3 x with phosphate-buffered saline (PBS). Coverslips were placed in 0.1% Triton X-100 (Avantor) for 10 min for permeabilization, followed by three PBS washes and blocking with 5% bovine serum albumin (BSA; Merck) for 1 hr. Coverslips were incubated at 4 °C overnight with anti-$P2Y_2$ (APR-010, Alomone labs) and anti-integrin αV antibodies (P2W7, Santa Cruz) diluted in blocking solution (1:100 and 1:200, respectively). After three PBS washes, coverslips were incubated for 1 hr with Alexa Fluor 647 goat anti-mouse and Alexa Fluor 488 goat anti-rabbit (Invitrogen) or Alexa Fluor 546 goat anti-rabbit at 1:1000, diluted in blocking buffer. Following three PBS washes, 4',6-diamidino-2-phenylindole (DAPI, Sigma-Aldrich) was used as a nuclear stain and was incubated for 10 min. Slides were mounted using Mowiol (Calbiochem) and imaged 24 hr later using a LSM 710 confocal microscope (Zeiss).

## siRNA and plasmid transfection

Cells were seeded in six well plates at a density of 200,000 cells/well 24 hr before transfection. For siRNA experiments, cells were transfected with 20 nM pooled control or $P2Y_2$-targeting siRNAs from a

siGENOME SMARTpool (Dharmacon, GE Healthcare) with Lipofectamine 3000 (Invitrogen) following the manufacturer's instructions. For P2Y$_2$ plasmid expression experiments, cells were transfected with 500 nM *P2RY2* (P2Y$_2^{RGD}$) or *P2RY2D97E* (P2Y$_2^{RGE}$) in pcDNA3.1 vector (Obtained from GenScript) or pcDNA3.1 alone (Empty vector, EV) together with lipofectamine 3000 and p3000 reagent (Invitrogen) as per manufacturer's instructions. Plasmid concentration was selected by comparing AsPC-1 IF staining of P2Y$_2$ with IF staining in AsPC-1 P2Y$_2^{CRISPR}$ and PANC-1 cells with different concentrations of the plasmid to achieve a similar IF signal. Cells were split 48 hr post-transfection for experiments or imaged 72 hr post-transfection.

### 3D sphere model invasion assay

Spheres of PDAC cell lines with PS-1 cells were generated as described (*Murray et al., 2022*). Cancer cells at 22,000 cells/mL and PS-1 cells at 44,000 cells/mL were combined with DMEM/F-12 and 1.2% methylcellulose in a 4:1 ratio of methylcellulose (Sigma-Aldrich) and 20 µl drops, each containing 1000 cells, pipetted on the underside of a 15 cm dish lid (Corning) and hanging drops were incubated overnight at 37 °C. The next day, spheres were collected and centrifuged at 300 g for 4 min and washed with the medium. A mix of 2 mg/mL collagen (Corning), 175 µL/mL Matrigel, 25 µL/mL HEPES (1 M, pH 7.5), and 1 N NaOH (for neutral pH correction) was prepared with DMEM/F12 medium. Spheroids were re-suspended and seeded in low attachment 96-well plates (50 µl per well) with 40 µL previously gelled mix in the bottom of the wells. Once set, 150 µL of DMEM/F12 was added with treatments. Spheres were treated with 100 µM adenosine 5'-triphosphate trisodium salt hydrate (ATP, Sigma), uridine 5'-triphosphate trisodium salt hydrate (UTP, Sigma) or adenosine 5'-[γ-thio]triphosphate tetralithium salt (ATPγS, Tocris) alone or with 5 µM AR-C118925XX (AR-C, Tocris), or 10 µM cyclo(RGDfV) (cRGDfV, Sigma-Aldrich). Treatments were repeated 24 hr later. Spheres were imaged with a Zeiss Axiovert 135 light microscope at 10 x on day two after seeding. Cells were stained with 4',6-diamidino-2-fenilindol (DAPI) (1:1000) for 10 min and imaged with a Zeiss LSM 710 confocal microscope. % Invasion was calculated by drawing an outline around the total area $A_{total}$ and central area $A_{central}$ of the spheres with ImageJ (Fiji) and using the equation:

$$\%Invasion = \left( \frac{A_{total} - A_{central}}{A_{central}} \right) x100$$

Results were plotted in SuperPlots by assigning different colors to repeats and superimposing a graph of the average % Invasion with a darker shade of the assigned color as described previously (*Lord et al., 2020*).

### IncuCyte migration assay

In IncuCyte ClearView 96-well cell migration plates (Essen BioScience), 40 µL medium with 5000 cells were seeded in each well. A solution of 20 µL medium with 15 µM AR-C or 30 µM cRGDfV was added on top of the wells to achieve a final concentration of 5 µM and 10 µM, respectively. Cells were allowed to settle for 15 min at room temperature and then placed at 37 °C for pre-incubation with the treatments for another 15 min. A volume of 200 µL of medium with or without 100 µM ATP was added in the appropriate reservoir wells and the plate was placed in the IncuCyte S3 (Essen BioScience) and was monitored every 4 hr for 39 hr (average doubling time of AsPC-1 cells *Chen et al., 1982*). Using the IncuCyte S3 2019 A software, the migration index was calculated by analyzing the average area occupied by the cells in the bottom well and was averaged with the initial average area occupied by cells in the top well.

### RNA extraction and qPCR analysis

RNA was extracted using the Monarch RNA extraction kit (New England BioLabs) as instructed by the manufacturer. The extracted RNA was quantified using a Nanodrop One Spectrophotometer (ThermoFisher Scientific). Using LunaScript RT Supermix kit (BioLabs), cDNA was prepared in a 20 µL reaction according to the manufacturer's instructions. The resulting cDNA was used in conjunction with MegaMix-Blue and *P2RY2* primers (Eurogentec; Forward sequence: GCTACAGGTGCCGCTTCAAC , reverse sequence: AGACACAGCCAGGTGGAACAT) (*Hu et al., 2019*) for quantitative polymerase chain reaction (qPCR) at the manufacturer's recommended settings in a StepOnePlus Real-Time PCR

System (Applied Biosystems). The relative mRNA expression was calculated using the $2^{-\Delta\Delta Ct}$ method (*Livak and Schmittgen, 2001*) and normalized to GAPDH.

## DNA-antibody coupling reaction

DNA labeling of anti-αV antibody (P2W7, Santa Cruz, RRID:AB_627116) and anti-P2Y$_2$ receptor antibody (APR-010, Alomone labs, RRID:AB_2040078) was performed via maleimidePEG2-succinimidyl ester coupling reaction as previously described (*Simoncelli et al., 2020*; *Joseph et al., 2021*). First, 30 µL of 250 mM DDT (Thermo Fisher Scientific) was added to 13 µL of 1 mM thiolated DNA sequences 5'-Thiol-AAACCACCACCACCA-3' (Docking 1), and 5-Thiol-TTTCCTCCTCCTCCT-3' (Docking 2) (Eurofins). The reduction reaction occurred under shaking conditions for 2 hrs. 30 min after the reduction of the thiol-DNA started, 175 µL of 0.8 mg/mL antibody solutions were incubated with 0.9 µL of 23.5 mM maleimide-PEG2-succinimidyl ester cross-linker solution (Sigma-Aldrich) on a shaker for 90 min at 4 °C in the dark. Prior to DNA-antibody conjugation, both sets of reactions were purified using Microspin Illustra G-25 columns (GE Healthcare) and Zeba spin desalting columns (7 K MWCO, Thermo Fisher Scientific), respectively, to remove excess reactants. Next, coupling of anti-P2Y$_2$ with DNA docking 1 and anti-αV with DNA Docking 2 was performed by mixing the respective flow-through of the columns and incubate them overnight, in the dark, at 4 °C under shaking. Excess DNA was removed via Amicon spin filtration (100 K, Merck) and antibody-DNA concentration was measured using a NanoDrop One spectrophotometer (Thermo Fisher Scientific) and adjusted to 10 µM with PBS. Likewise, spectrophotometric analysis was performed to quantify the DNA-antibody coupling ratio and found to be ~1.2 on average for both the oligo-coupled primary antibodies.

## Cell fixation and immunofluorescence staining for DNA-PAINT imaging

Cells were seeded at 30,000 cells per channel on a six-channel glass-bottomed microscopy chamber (µ-SlideVI$^{0.5}$, Ibidi) pre-coated with rat tail collagen type I (Corning). The chamber was incubated at 37 °C for 8 hr before treatments. Cells were treated with 100 µM of ATP (or the equivalent volume of PBS as control) in the medium for 1 hr and were fixed and permeabilized as described in the 'Cell fixation and immunofluorescent staining' section. Following permeabilization, samples were treated with 50 mM ammonium chloride solution (Avantor) for 5–10 min to quench auto-fluorescence and cells were washed 3x in PBS. Blocking was completed via incubation with 5% BSA (Merck) solution for 1 hr followed by overnight incubation at 4 °C with 1:100 dilutions of DNA labeled anti-P2Y$_2$, and DNA labeled anti-αV antibody in blocking solution. The next day, samples were washed 3× in PBS and 150 nm gold nanoparticles (Sigma-Aldrich) were added for 15 min to act as fiducial markers for drift correction, excess of nanoparticles were removed by 3x washes with PBS. Samples were then left in DNA-PAINT imager buffer solution, prepared as described below, and immediately used for DNA-PAINT imaging experiments.

## DNA-PAINT imager solutions

A 0.1 nM P2Y$_2$ imager strand buffer solution (5-TTGTGGT-3'-Atto643, Eurofins) and a 0.2 nM αV imager strand buffer solution (5-GGAGGA-3'-Atto643, Eurofins) were made using 1x PCA (Sigma-Aldrich), 1x PCD (Sigma-Aldrich), 1x Trolox (Sigma-Aldrich), 1x PBS, and 500 mM NaCl (Merck) which facilitates the establishment of an oxygen scavenging and triplet state quencher system. Solutions were incubated for 1 hr in the dark before use. Stock solutions of PCA, PCD, and Trolox were prepared as follows: 40x PCA (protocatechuic acid) stock was made from 154 mg of PCA (Sigma-Aldrich) in 10 mL of Ultrapure Distilled water (Invitrogen) adjusted to pH 9.0 with NaOH (Avantor, Radnor Township, PA, USA). 100 x PCD (protocatechuate 3,4-dioxygenase) solution was made by adding 2.2 mg of PCD (Sigma-Aldrich) to 3.4 mL of 50% glycerol (Sigma-Aldrich) with 50 mM KCl (Sigma-Aldrich), 1 mM EDTA (Invitrogen), and 100 mM Tris buffer (Avantor). 100 x Trolox solution was made by dissolving 100 mg of Trolox in 0.43 mL methanol (Sigma-Aldrich), 0.345 mL 1 M NaOH, and 3.2 mL of Ultrapure Distilled water.

## Exchange-PAINT Imaging Experiments

Exchange DNA-PAINT imaging was performed on a custom-built total internal reflection fluorescence (TIRF) microscope based on a Nikon Eclipse Ti-2 microscope (Nikon Instruments) equipped with a 100×oil immersion TIRF objective (Apo TIRF, NA 1.49) and a Perfect Focus System. Samples were

imaged under flat-top TIRF illumination with a 647 nm laser (Coherent OBIS LX, 120 mW), that was magnified with custom-built telescopes, before passing through a beam shaper device (piShaper 6_6_VIS, AdlOptica) to transform the Gaussian profile of the beam into a collimated flat-top profile. The beam was focused into the back focal plane of the microscope objective using a suitable lens (AC508-300-A-ML, Thorlabs), passed through a clean-up filter (FF01-390/482/563/640-25, Semrock), and coupled into the objective using a beam splitter (Di03-R405/488/561/635-t1-25 × 36, Semrock). Laser polarization was adjusted to circular after the objective. Fluorescence light was spectrally filtered with an emission filter (FF01-446/523/600/677-25, Semrock) and imaged on an sCMOS camera (ORCA-Flash4.0 V3 Digital, Hamamatsu) without further magnification, resulting in a final pixel size of 130 nm in the focal plane, after 2 × 2 binning. For fluid exchange, each individual chamber of the ibidi μ-SlideVI$^{0.5}$ was fitted with elbow Luer connector male adaptors (Ibidi) and 0.5 mm silicon tubing (Ibidi). Each imaging acquisition step was performed by adding the corresponding imager strand buffer solution to the sample. Prior to the imager exchange, the chamber was washed for 10 min with 1 x PBS buffer with 500 mM NaCl. Before the next imager strand buffer solution was added, we monitored with the camera to ensure the complete removal of the first imager strand. Sequential imaging and washing steps were repeated for every cell imaged. For each imaging step, 15,000 frames were acquired with 100ms integration time and a laser power density at the sample of 0.5 kW/cm$^2$.

## Super-resolution DNA-PAINT image reconstruction

Both P2Y$_2$ and αV Images were processed and reconstructed using the Picasso (*Schnitzbauer et al., 2017*) software (Version 0.3.3). The Picasso 'Localize' module was used to identify and localize the x, y molecular coordinates of single molecule events from the raw fluorescent DNA-PAINT images. Drift correction and multi-color data alignment were performed via the Picasso 'Render' module, using a combination of fiducial markers and multiple rounds of image sub-stack cross-correlation analysis. Localizations with uncertainties greater than 13 nm were removed and no merging was performed for molecules re-appearing in subsequent frames. Super-resolution image rendering was performed by plotting each localization as a Gaussian function with a standard deviation equal to its localization precision.

## Protein quantification via qPAINT analysis

To convert the list of *x, y* localizations into a list of *x, y* protein coordinates the data was further processed using a combination of DBSCAN cluster analysis, qPAINT analysis, and *k*-means clustering.

First, 21 randomly selected, non-overlapping, 4 × 4 μm$^2$ regions of interest (ROIs) for each type of cell and cell treatment were analyzed with a density-based clustering algorithm, known as DBSCAN. To avoid suboptimal clustering results; ROIs were selected such that they do not intersect with cell boundaries and the regions were the same for P2Y$_2$ and αV images. Single-molecule localizations within each ROIs were grouped into clusters using the DBSCAN modality from PALMsiever (*Pengo et al., 2015*) in MATLAB (Version 2021a)(*Pengo et al., 2015*). This clustering algorithm determines clusters based on two parameters. The first parameter is the minimum number of points ('minPts') within a given circle. For minPts, we chose a parameter in accordance with the binding frequency of the imager strand and acquisition frame number; in our case this was set to 10 localizations for all the experiments. The second parameter is the radius (epsilon or 'eps') of the circle of the cluster of single molecule localizations. This is determined by the localization precision of the super-resolved images and, according to the nearest neighbor based analysis was ca. to 10 nm for all the images.

For qPAINT analysis, we used a custom-written MATLAB (Version 2021a) code: https://github.com/Simoncelli-lab/qPAINT_pipeline (*Joseph and Simoncelli, 2023*). Briefly, localizations corresponding to the same cluster were grouped and their time stamps were used to compile the sequence of dark times per cluster. All the dark times per cluster were pooled and used to obtain a normalized cumulative histogram of the dark times which was then fitted with the exponential function $1 - \exp(t/\tau_d)$ to estimate the mean dark time, $\tau_d$, per cluster. The qPAINT index ($q_i$) of each cluster was then calculated as the inverse of the mean dark time, $1/\tau_d$.

Calibration was then performed via a compilation of all qPAINT indexes obtained from the DNA-PAINT data acquired for each protein type into a single histogram. Only qPAINT indices corresponding to small clusters (i.e. clusters with a maximum point distance of 150 nm) were considered.

This histogram was fitted with a multi-peak Gaussian function to determine the qPAINT index for a cluster of single molecule localizations corresponding to one protein ($q_{i1}$).

The calibration value obtained with this method was used to estimate the number of P2Y$_2$ and αV proteins in all the single-molecule localizations clusters identified by DBSCAN, as this corresponds to the ratio between $q_{i1}$ and the qPAINT index of each cluster. Finally, $k$-means clustering was used to recover a likely distribution of the proteins' positions in each cluster of single molecule localizations, where $k$ is equal to the number of proteins in that cluster. This information allowed us to quantify the protein density and level of protein clustering.

## Nearest neighbor analysis

Nearest neighbor distances (NND) for P2Y$_2$ – P2Y$_2$ and αV-αV were calculated using the recovered P2Y$_2$ and αV-protein maps as described above via a custom-written MATLAB (Version 2021a) script: https://github.com/Simoncelli-lab/qPAINT_pipeline (*Joseph and Simoncelli, 2023*). For colocalization analysis, the NND for each protein of one dataset with respect to the reference dataset was calculated (i.e. P2Y$_2$ - αV) using a similar MATLAB script. To evaluate the significance of the NND distributions, we randomized the positions of P2Y$_2$ and αV for the comparison of P2Y$_2$ – P2Y$_2$ and αV-αV NND distributions, respectively, and the positions of one of the two proteins for the comparison of the NND between P2Y$_2$ - αV protein distributions. The resulting histogram of the nearest neighbor distances for both the experimental data sets and the randomly distributed data was normalized using the total number of NND calculated per ROI to calculate the percentage of the population with distances smaller than a set threshold value.

## Western blotting

Cell lysates were extracted using RIPA buffer and 20 μg denatured protein per sample was loaded and separated using an 8% SDS-PAGE gel. Gels were run at 150 V for 2 hr and transferred into a nitrocellulose membrane (GE Healthcare) at 100 V for 1 hr. Following blocking with 5% milk (Sigma) in 0.1% TBS-T for 1 hr, membranes were incubated with 1:1000 dilution of antibodies against phosphorylated FAK (Tyr397, 3283, Cell Signaling, RRID:AB_2173659), phosphorylated ERK 1/2 (S217/221, 9154, Cell Signaling, RRID:AB_2138017), P2Y$_2$ (APR-010, Alomone Labs, RRID:AB_2040078), HSC 70 (SC7298, Santa Cruz, RRID:AB_627761), or α-tubulin (T5168, Sigma-Aldrich, RRID:AB_477579) with 5% BSA in 0.1% TBS-T overnight at 4 °C. Membranes were probed with anti-Mouse-HRP (P0447, DAKO, RRID:AB_2617137), or Anti-Rabbit-HRP (P0448, DAKO, RRID:AB_2617138) at 1:5000 in 5% milk in TBS-T for 1 hr at room temperature. Images were captured by using Luminata Forte Western HRP substrate (Millipore) and imaged with an Amersham Imager 600 (GE Healthcare).

## Statistical analysis

For the statistical analysis of the number and colocalization of DNA-PAINT images, a minimum of five $4 \times 4$ μm$^2$ regions obtained from AsPC-1 cells were analyzed per condition. For all experiments, normality tests were performed and the non-parametric Kruskal-Wallis test for significance was calculated. All graphs and statistical calculations of experimental data were made using Prism 9.4.1 (GraphPad).

## Acknowledgements

We thank the Barts Pancreatic Tissue Bank (BPTB) for providing pancreatic tissue slides presented in this work. BPTB is supported by Pancreatic Cancer Research Fund and we thank all its members, in particular, Prof. Claude Chelala, Christine Hughes, Ahmet Imrali, and Amina Hughes for help, as well as Consultant Pathologist Dr. Joanne Chin-Aleong and members of Tissue Access Committee and Operations Group. We thank Dr. Ann-Marie Baker for her expertise in RNAscope experiments. This work was supported by a Medical Research Council (MRC) iCase award to PJM and RPG from Barts Charity and the MRC Doctoral Training Programme for ETB at Queen Mary University of London (Project MRC0227). NJR acknowledges the QMUL MRC Doctoral Training Program (MR/N014308/1). MDJ acknowledges support from the BBSRC (BB/T008709/1) via the London Interdisciplinary Doctoral Programme and SS acknowledges financial support from the Royal Society through a Dorothy Hodgkin fellowship (DHF\R1\191019) and a Research Grant (RGS\R2\202038). This work was supported by

Cancer Research UK (CRUK) awarded to EPC and RPG (A27781) and a CRUK Centre grant to Barts Cancer Institute (A25137). Diagrams were generated using BioRender.

## Additional information

### Funding

| Funder | Grant reference number | Author |
| --- | --- | --- |
| Cancer Research UK | A27781 | Edward P Carter |
| Cancer Research UK | A25137 | Edward P Carter Richard P Grose |
| Medical Research Council | MRC0227 | Elena Tomas Bort |
| Biotechnology and Biological Sciences Research Council | BB/T008709/1 | Megan D Joseph |
| Royal Society | DHF\R1\191019 | Sabrina Simoncelli |
| Royal Society | RGS\R2\202038 | Sabrina Simoncelli |
| Medical Research Council | MR/N014308/1 | Nicolas J Roth |
| Pancreatic Cancer Research Fund | Tissue Bank Grant | Hemant M Kocher |

The funders had no role in study design, data collection and interpretation, or the decision to submit the work for publication.

### Author contributions

Elena Tomas Bort, Conceptualization, Software, Formal analysis, Investigation, Visualization, Writing – original draft, Writing – review and editing; Megan D Joseph, Software, Formal analysis, Investigation, Writing – original draft; Qiaoying Wang, Jessica Gibson, Formal analysis, Investigation; Edward P Carter, Investigation, Methodology; Nicolas J Roth, Ariana Samadi, Investigation; Hemant M Kocher, Resources, Writing – review and editing; Sabrina Simoncelli, Software, Methodology, Writing – original draft, Writing – review and editing; Peter J McCormick, Conceptualization, Supervision, Funding acquisition, Project administration, Writing – review and editing; Richard P Grose, Conceptualization, Supervision, Funding acquisition, Investigation, Project administration, Writing – review and editing

### Author ORCIDs

Elena Tomas Bort  http://orcid.org/0000-0001-7897-8891
Megan D Joseph  http://orcid.org/0000-0001-8313-8467
Edward P Carter  http://orcid.org/0000-0003-4499-1101
Sabrina Simoncelli  http://orcid.org/0000-0001-7089-7667
Peter J McCormick  http://orcid.org/0000-0002-2225-5181
Richard P Grose  http://orcid.org/0000-0002-4738-0173

### Decision letter and Author response

Decision letter https://doi.org/10.7554/eLife.86971.sa1
Author response https://doi.org/10.7554/eLife.86971.sa2

## Additional files

### Supplementary files

• Supplementary file 1. Pancreatic cancer molecular subtypes associated with purinergic gene expressions. Purinergic genes with significantly higher expression in a specific molecular subtype have been listed below. If no significant higher expression was observed not applicable (N/A) is shown.

• MDAR checklist

## Data availability

All data generated or analysed during this study are included in the manuscript and supporting file, or online resources are fully referenced. Human PDAC tumour data were generated by TCGA Research Network (https://www.cancer.gov/tcga) and by the Clinical Proteomic Tumour Analysis Consortium (https://www.proteomics.cancer.gov). The Genotype-Tissue Expression (GTEx) Project was used for the analysis of normal pancreatic tissue samples (https://gtexportal.org).

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
