## [Editor Report]

In this manuscript, the authors address an important and urgent question: what molecular mechanisms drive the invasive behavior of pancreatic adenocarcinoma? Because these tumors have such a strong propensity for invasion and metastasis, identifying actionable targets is of high importance. Using a combination of in silico and in vitro modeling, they identify a role for purinergic G-protein coupled receptor P2Y_2_ as a critical node in mediating PDAC invasion, and they find that blocking the crosstalk between P2Y_2_ and αV integrins via a peptide inhibitor blocks PDAC invasion, which may have clinical utility. Thus their study provides insights into both the basic biology of PDAC but also identifies a new target.

---

## [Decision Letter]

[Editors' note: this paper was reviewed by Review Commons.]

---

## [Author Response]

1. General Statements [optional]

We are grateful to the reviewers for highlighting the novelty of the mechanism we describe for P2Y2 in driving RGD-binding integrin-dependent invasion, and acknowledging its potential in cancer therapy. We thank the reviewers for their valuable and detailed comments, which have allowed us to prepare a significantly stronger and clearer manuscript.

2. Point-by-point description of the revisionsReviewer #1 (Evidence, reproducibility and clarity (Required)):SummaryThe study identifies P2Y2 as a purinergic receptor strongly associated with hypoxia, cancer expression and survival. A link is found between P2Y2-integrin interaction and cancer invasion, highlighting this as a novel therapeutic target. The mechanism is interesting and general well explored.

We thank the reviewer for acknowledging the novelty of the therapeutic target presented in this work.

Minor commentsAs P2Y2 is highly expressed by other cell types found with tumours, including vascular endothelium and leukocytes, the authors should reflect on this as a confounding factor in the analysis of adrenocarcinoma gene expression analysis. I appreciate the RNAscope work may resolve this issue to some extent.

We agree that P2Y2 is known to be expressed in other cell types. RNAscope did not show convincing staining in PDAC normal adjacent tissue (was similar to negative staining), perhaps due to the challenging nature of pancreatic tissue with respect to RNA degradation. We have resolved this issue by including single cell RNA-seq of normal human pancreas for P2Y2 from Protein Atlas (Sup. Figure 2B), which shows expression in several cell types, mainly endocrine cells, and macrophages. We now mention this in line 142 : “P2Y2 is known to be expressed at low levels in normal tissues but interestingly RNAscope did not detect this. This data suggest (1) the lower limits of the technique compounded by the challenge of RNA degradation in pancreatic tissue and (2) supports that in tumour tissue where it was detected there was indeed overexpression of P2Y2, in line with the bioinformatic data. Interrogating single cell P2Y2 RNA expression in normal PDAC from proteinatlas.org (Karlsson et al., 2021), expression was found at low levels in several cells types, for example in endocrine cells and macrophages (Sup. Figure 2B).”

Major commentsThe authors correctly identify that the level of ATP in the tumour microenvironment can be very high, typically 100uM or so. However, these concentrations are supramaximal for P2Y2 activation, at which ATP has an approximate EC50 of 100nM. Coupled with the fact that many cell types, including cancer cells, constitutively secrete ATP, there is an opportunity to explore the effects of lower ATP concentrations in some assays, or provide some concentration-response relationship to give more confidence of P2Y2-dependent effects.

We thank the reviewer for raising this point and we agree that 100 μm can be a high concentration, albeit one that is frequently used throughout the literature. We have now included a concentration-response relationship (Sup. Figure 2D) showing that ATP causes cytoskeletal changes that are P2Y2 dependent most prominently at 100 uM, the concentration that, as the reviewer has also corroborated, is similar to the concentration of ATP found in tumours.

Also, the authors describe the use of cancer cells where P2Y2 has been knocked out using CRISPR. Does this KO have an effect on cancer invasion? The effect of ARC should be absent in these cells and give confidence the effects of ARC are P2Y2-dependent, as some off-target effects of this antagonist have been reported. To explore the influence of constitutive P2Y2 activity, the authors should explore the effects of ARC alone in some assays.

We agree that including more AR-C only experiments would be informative, so we have included a 3D sphere invasion assay with our CRISPR cell line treated with and without AR-C that shows no effect in invasion (*p* = 0.4413) (Sup. Figure 3J). We have now also included images of AsPC-1 cells transfected with Lifeact, showing no changes in morphology with AR-C only (Sup. Figure 2E). We apologise for missing a ‘+’ in one of the supplementary figures which shows AR-C only in AsPC-1 cells has no effect on its own.

The effects of the CRISPR cell line in invasion are shown in Figure 3F, showing a significant reduction (*p* = 0.0005) in invasion.

The title of the manuscript implies extracellular ATP drives cancer invasion, though in my opinion this statement is not fully explored. Though ATP/UTP are applied at supramaximal concentrations for P2Y2 activation, the influence of ATP in the cell culture microenvironment without exogenous application is not explored. One would predict that scavenging extracellular ATP with apyrase would negatively impact invasiveness and the proximity of integrin and P2Y2 without ATP/UTP application if constitutively secreted ATP is involved. Pharmacological manipulation of ectonucleotidase activity is an alternative. Experimental route to explore this.

We agree and have changed the title of our article to “Purinergic GPCR-integrin interactions drive pancreatic cancer cell invasion”. Our 3D sphere experiments with the CRISPR cell line show a reduction in invasion without exogenous application of ATP, which we also see to a lesser extent in our siRNA P2Y2 cell line. We have tested our sphere model with apyrase but unfortunately, the buffer used for apyrase to work is not compatible with our gel composition. Pharmacological manipulation would be a very good alternative if the cells used expressed high levels of CD39 or PANX1, which unfortunately they don’t. We hypothesise that most basal extracellular ATP in our 3D spheres comes from hypoxic areas that cause cell death, just as is postulated for tumours.

Immunoprecipitation experiments of native proteins would be more convincing data that P2Y2 and integrin physically interaction, as opposed to being in close proximity. This would also overcome artifacts of interaction that can be attributed to receptor overexpression.

We attempted immunoprecipitation experiments but unfortunately ran into several technical difficulties, including the anti-αV antibody working poorly for Western blot. Immunoprecipitation of these proteins has been reported by others (PMID: 25908848), supporting the proposed interaction.

DNA-PAINT super resolution microscopy allows for quantification of nanoscale distances, and we used this to calculate the distances where physical interaction occurs. The possibility of this close proximity being by chance is accounted for in the computational nearest neighbour distance calculation by calculating points randomly distributed. This random distribution calculation also helps in overcoming artifacts of interaction due to overexpression, as the random distributed points are the same number of points as the proteins detected in each condition for each region of interest. Importantly, we also performed DNA-PAINT in using untransfected AsPC-1 thus endogenous levels (no receptor overexpression or alteration) and saw similar results (Sup. Figure 4A-D), thus we are confident of the interactions reported.

Finally, we alter the RGD motif, which underpins the physical interaction, and see significant changes that match observations in previous publications using the P2Y2 agonist UTP, mentioned in the discussion: Line 398 “Following ATP stimulation, the number of P2Y2 proteins at the plasma membrane decreased significantly after one hour, implying receptor internalisation, in line with previous work showing P2Y2 at the cell surface was reduced significantly after one hour of UTP stimulation (Tulapurkar et al., 2005).” and Line 408: “P2Y2 affecting cell surface redistribution of αV integrin has been reported, with αV integrin clusters observed after 5 min stimulation with UTP (Chorna *et al.*, 2007)”

It is currently not clear what the mechanistic relationship between P2Y2 activity, P2Y2-integrin proximity and RGD motif is. Do the authors suggest the RGD domain becomes exposed upon receptor activation? The mechanism is not fully articulated in the discussion.

We apologise for any lack of clarity in our postulated mechanism, we have now included a more detailed explanation of the mechanism in the discussion : Line 417 “We speculate that by reducing the ability of integrins to bind to the RGD of P2Y2, through receptor internalisation, RGE mutation or through cRGDfV treatment, there is less RGD-triggered integrin endocytosis, hence less integrin recycling and an increase of integrins at the cell surface.”

Reviewer #1 (Significance (Required)):General assessment: A novel mechanism is presented for therapeutic intervention of cancer. The study relies on supramaximal concentrations of agonist and overexpressed receptors. Role of endogenous P2Y2 not fully explored. The study lacks in vivo evidence of the importance of this mechanisms. Cell developed in the study could be used in mouse models to explore effect on tumour growth.Advance: Integrin and P2Y2 interactions are already documented but not in context of cancer.Audience: basic research

We thank the reviewer for crediting this work as a novel mechanism for therapeutic intervention of cancer. We trust that the new data provided (as discussed above) have resolved the concerns of the reviewer as we now have provided an explanation for the concentrations used. We do rely on overexpressed receptors for a small portion of our experiments, however, all experiments with overexpressed receptors were then tested in cells with endogenous expression of P2Y2 and used pharmacological means to show the same behaviour. We have now clarified this. We have also included in the discussion a sentence about the mouse experiment performed by Hui et al. with regards to reduced tumour growth when targeting P2Y2: Line 365: “Combination treatment of subcutaneous xenografts of AsPC-1 or BxPC-3 cells with the P2Y2 antagonist AR-C together with gemcitabine significantly decreased tumour weight and resulted in increased survival compared to placebo or gemcitabine monotherapy control (Hu et al., 2019).”

Reviewer #2 (Evidence, reproducibility and clarity (Required)):Summary:Considering the fact that most PDAC are characterized by a high level of extracellular purines content, authors decided to study the expression of the 23 genes coding for membrane proteins involved in the binding or transport of purines in available PDAC transcriptomic cohorts. This approach led to the identification of P2Y2, a GPCR, as the best predictor for the worst survival of patients. Using in vitro models, they show that P2Y2 expression is associated with increased invasion capacity of pancreatic cancer cells and that this pro-invasive effect is dependent on the interaction of P2Y2 with αV integrin via the RGD motif.Major comments:It is not clear to me why authors decided at one point to perform a GSEA comparing low and high mRNA expression of P2Y2 and why they decided to focus on the potential interaction of P2Y2 with integrin αV. As a GPCR, activation of P2Y2 leads to the activation of several downstream signaling pathways that may directly impact the adhesion, migration, and invasion properties of cells. Moreover, despite the presence of the RGD motif in P2Y2, it is not excluded that it may bind (maybe more efficiently) to other "cell adhesion" molecules.

We apologise if the link between the GSEA figure and focusing on the potential integrin interaction was not clear. We have now performed GSEA using the panther gene set library, which includes a “Integrin signalling pathway” gene set. This was the top ranked gene set in both cohorts and we have substituted the GSEA figure for this instead (Figure 2D). We trust that the narrative of the manuscript and our rationale to pursue the importance of integrin interaction is now clear.

We agree with the reviewer and believe that P2Y2 may bind to other molecules important in cell adhesion. We studied integrin interactions due to the clear relationship of P2Y2 and integrins in patient data, which was not as evident with other binding partners. Furthermore, this relationship is unexplored in cancer and offers novel therapeutic strategies.

Similarly, if αV can regulate P2Y2 signaling, what about the regulation of αV signaling pathways by P2Y2? αV integrin has to bind to a β subunit and, depending on the identity of the β subunit, may have distinct regulations and so different impact on cell invasion. How P2Y2 can interfere with these α/β ratios?

We thank the reviewer for this comment, and have now included western blots showing the impact of P2Y2 treatment on integrin signalling through FAK and ERK (Figure 5). We agree that the β subunit may have distinct regulation and outputs, but this is outwith the scope of our current study.

While it has been shown in other studies, in this work, there is no real proof of the interaction between P2Y2 and αV. Only in Figure 4I, where the authors look at the NND <20nm between both proteins, we can see that only 1 to 2 % of αV is in close proximity with P2Y2, which seems very low.

We thank the reviewer for raising this point as it has made us realise that our chosen NND of <20 nm, in an attempt of being cautious, consistant, and only select true physical interaction was too strict for some conditions. We have adjusted this to the maximum NND required for the proteins to physically touch based on individual protein and antibody dimensions (Sup. Figure 5A). The resulting changes are <50 nm for P2Y2-αV, <20 nm for αV-αV and <40 nm for P2Y2-P2Y2, giving more realistic percentages for each condition. We then verified this range using the NND histograms now included in the manuscript, which additionally provide information about the distribution of the NNDs in each protein-protein dynamic (Sup. Figure 5B). One important point is that integrins are part of a large number of protein complexes and interact with many proteins, hence only a small percentage will be interacting with P2Y2.

Surprisingly, in the absence of ATP, P2Y2 RGE mutant, which should no more interact with αV, show a 2 to 3 fold more vicinity to αV compared to WT P2Y2. How can the authors explain this?

We agree that this is a surprising, but robust discovery. By altering the RGD motif, there may be less RGD-triggered integrin endocytosis, leading to increased integrins at the surface. We have included this hypothesis in the discussion in Line 417. The RGE mutation has less affinity to integrins, meaning it still retains some ability to bind to integrins. Hence by chance, a higher number of integrins will result in a higher number of interactions with the RGE. We are planning to interrogate the internalisation dynamics in a future study.

For DNA-PAINT experiments, the authors only focus on membrane proteins whose amounts are balanced by internalization, recycling and export from internal compartment. As claimed, but not demonstrated by the authors, interaction of P2Y2 and αV may interfere with all these steps, thereby increasing or decreasing the cell surface expression of both proteins. Hence, it would be useful to (1) control proteins levels by western blot, especially for the overexpressed P2Y2, to be sure that they are the same, (2) block internalization and/or export to decipher the important steps.In fact, all these main questions are raised by the authors in the end of the discussion but so far, they only show that the RGD motif has an impact on the biological role of P2Y2 (cell invasion) and on the membrane dynamic of αV and itself.

We thank the reviewer for the suggestions:

1) In the course of our attempts to perform co-IP for P2Y2 and αV we could confirm that P2Y2 expression levels were equivalent (see Author response image 1), but the problems with anti-αV antibodies prevented completion of the experiment. We also show IF staining showing similar levels of P2Y2 for both overexpressed conditions (Sup. Figure 3K).

**Author response image 1. sa2fig1:** Immunoprecipitation of P2Y_2_ showing similar P2Y_2_ levels in AsPC-1 P2Y_2_^CRISPR^ cells transfected with P2Y_2_^RGD^ or P2Y_2_^RGE^ and treated with 100 µM of ATP or control for 1 hour. Antibody used: anti-P2Y_2_ (APR-010, Alomone Labs).

2) As the reviewer highlights, in this work we have focused on the role of P2Y2 in PDAC invasion and have looked at single-molecule resolution membrane dynamics of αV and P2Y2. The different steps of P2Y2 and integrin αV interactions in internalisation, recycling and export are certainly interesting to study but beyond the scope of the current manuscript and in our future aims. We include these ideas in the discussion as suggestions for future research and as a possible explanation for the dynamics observed.

Figure 2A, authors use RNAscope in order to reveal P2Y2 mRNA expression and distribution in tumor versus normal tissue from 2 patients. They rather show the protein expression, using the antibody they used in other experiments, by standard IHC and in a higher number of patients, including short and long survival, to confirm that the results they obtain by bioinformatics study of transcriptomic data are real.

We now explicitly mention a paper (PMID: 30420446) that performed IHC of P2Y2 in 264 patients showing that P2Y2 was predominantly found in the tumour area, matching our bioinformatics study: Line 141 “matching our findings from larger publicly available cohorts, including P2Y2 IHC data from 264 patients in the Renji cohort (Hu et al., 2019).” and Line 359 “These observations were supported by published immunohistochemical staining of 264 human PDAC samples, showing that P2Y2 localised predominantly in cancer cells in human PDAC…”

Some figure legends are incorrectly numbered or described, such as the figure 4.

We apologise for the incorrectly described figures in figure 4, this has now been corrected.

Minor comments:Can we reasonably talk about OMIC while studying 23 genes? In fact, as described by Timothy A. J. Haystead in 2006 (PMID: 16842150) the purinome is constituted of about 2000 genes coding for proteins binding to purines (including all kinases for example). Author should redefine they pool of genes as perhaps purines receptors/transporter?

We agree with the reviewer and have redefined the pool of genes to ‘purinergic signalling genes’ or ‘(part of the) extracellular purinome’.

P2Y2 and ADORA2B associated with worse survival while P2Y11 and ADORA2A are associated with better survival (Figure 1B). Would it be more interesting to understand why proteins of the same family act in opposite ways?

We have now included text exploring this idea in the discussion. Both P2Y2 and ADORA2B show increased expression with HIF-1α and/or hypoxia and the inverse happens with ADORA2A, for example. Line 352: “Adenosine A2B receptor requires larger agonist concentrations for activation compared to other receptors in the same family, such as adenosine A2A (Bruns, Lu and Pugsley, 1986; Xing et al., 2016), and receptor expression has been reported to increase when cells are subjected to hypoxia (Feoktistov et al., 2004). Moreover, HIF-1α has been shown to upregulate A2B and P2Y2 expression in liver cancer (Tak et al., 2016; Kwon et al., 2019).”

Figure 1C, any value for the correlation with Survival? Cause this is not so obvious in the figure.

We agree this correlation needs strengthening with a numeric value, we have now included a Kaplan-Meier curve of high vs low Winter hypoxia score PDAC patients showing significantly lower survival with higher Winter hypoxia score (Sup. Figure 1B).

Regarding the correlation of P2Y2 and ADORA2B with hypoxia scores, any HIF1 responsive element in promoter? What happens regarding the expression level of these genes when cells are transferred to low oxygen conditions?

We thank the reviewer for these questions. The relationship of P2Y2 and ADORA2B with hypoxia and/or HIF-1α has been explored in other publications which are now cited in the discussion. Line 356: “Moreover, HIF-1α has been shown to upregulate A2B and P2Y2 expression in liver cancer (Tak *et al.*, 2016; Kwon *et al.*, 2019).” Of note, a HIF1-α responsive element has been reported for A2B, but as yet not for P2Y2.

Figure 4 E to M are too small.

We apologise and have now increased the size of the graphs and the figure.

In Supp Figure 4, what are the "Non-altered AsPC-1 cells"?

We apologise for the confusion that may have arisen from calling normal AsPC-1 cells “Non-altered AsPC-1 cells”. We have changed this to ‘Normal AsPC-1 cells (untransfected and unchanged P2Y2 expression).

Reviewer #2 (Significance (Required)):Strengths: All the data shown are experimentally and statistically strong.Limitations: This study remains largely descriptive with no real molecular mechanism that could at least partially explain the biological role of P2Y2 regarding cell invasion.Advance: Limited

We thank the reviewer for noting the experimental strength of the paper.

After the suggested changes, including integrin signalling experiments, and strengthening our DNA-PAINT results, the molecular mechanism presented in this work has been strengthened and clarified significantly. These changes have also helped greatly in the mechanistic explanation of the role of P2Y2 in cell invasion.

Reviewer #3 (Evidence, reproducibility and clarity (Required)):The authors concentrate on the members of the purinome and attempt to identify members of the pathway that are especially relevant for PDAC biology, especially invasion and metastatic spread. Using the in silico analysis of transcriptome data from publicly available PDAC patient cohorts, the authors identify P2Y2 as being the most prominent in terms of cancer cell expression and with highest impact on patient survival. The authors than take an effort in functional characterization of P2Y2 and demonstrate that downregulation/deletion of P2Y2 leads to abrogation of ATP activated invasion in hanging drop spheroid model system in a very reasonable and scientifically good way. Finally, the authors postulate that the P2Y2 actions go over interaction with integrin AlphaV and modulations of the cellular cytoskeleton and show via DNA PAINT that a direct interaction of the 2 molecules. The hypothesis is experimentally elaborated in a sound way mostly using cell culture as a system.The study is solid communicated, the number of experiments seems to be fine. For my understanding, the study relies much on mRNA data (gene expression in cell lines and patient samples), I would suggest providing evidence on protein level what might have been challenging due to potential lack of specific antibody.

We thank the reviewer for acknowledging our experimentally elaborated hypothesis and our solid communication of the study. As mentioned before, we now explicitly mention a paper (PMID: 30420446) that performed IHC of P2Y2 in 264 patients showing that P2Y2 was predominantly found in the tumour area, matching our bioinformatics study.

Reviewer #3 (Significance (Required)):To strengthen the hypothesis experimentally, I would suggest the experiments listed below:Figure 1: The authors took a solid bioinformatic effort and analyzed expression of different genes of the purinome pathway in different PDAC patient and cell gene expression databases. In this part, the authors rely a lot on correlation of hypoxia and define high hypoxia scores and low hypoxia scores from previously published datasets. Although hypoxia surely plays an important biological condition in the PDAC, I am not sure I get the connection between purinome pathway and hypoxia. Few sentences give a broad introduction about hypoxia-purinome connection in the discussion part of the manuscript, but I think the readership would benefit from more specific statements (which drug, which hypoxic target, which system-mouse/human/cells, what was the exact discovery) and connect those specific statements to the work that has been done here.

We agree with the reviewer that the study can benefit from more information about the hypoxia-purinergic signalling link. Hence, we have now included more detailed explanations of how hypoxia and purinergic signalling are related in the discussion, giving more information about the cell types and the exact discovery. Line 338: “Purinergic signalling has been associated classically with hypoxia and immune function in cancer (Di Virgilio et al., 2018). One of the first reports of hypoxia inducing ATP release in cells identified an increase of extracellular ATP in rat heart cells when kept in hypoxic conditions (Forrester and Williams, 1977). PDAC is a highly hypoxic cancer, with high levels of ATP reported in the tumour interstitial fluid of human and mouse PDAC tissues compared to healthy tissues (Hu et al., 2019).”

Do the authors attempt to state here that hypoxic PDACs are those with worse prognosis and more aggressive and thus try to associate members of the purine pathway with those "worse" PDACs? Surprisingly, there is relatively little knowledge about hypoxia in PDAC and I would not suggest using it in this context as a predictor. Reports do suggest that hypoxia forces the emerging of resistant phenotypes but if the authors want to use hypoxic signatures, they have to fortify better (with literature) why do they choose hypoxia and what is the hypothesis that connects hypoxia to purinome, what makes this connection worth investigating.

We thank the reviewer for raising the question of PDAC and worse prognosis with hypoxia. We have now included a Kaplan-Meier curve of high vs low Winter hypoxia score PDAC patients showing significantly lower survival with higher Winter hypoxia score (Sup. Figure 1B). The significant link with poor survival shown with hypoxia and the inclusion of more detailed explanation of the links with hypoxia and purinergic signalling proteins (mentioned above), now clarify the reasoning for investigating this connection.

I find the statement "hypoxia in tumor core" a bit tricky, acute and chronic hypoxia can occur anywhere in the tumor, to my knowledge there are no reports saying only the tumor core suffers from hypoxia in PDAC. PDAC being especially rich in stroma in all of its parts is probably more prone to overall hypoxia and not only in tumor core.

We agree that “hypoxia in tumour core” can be a tricky statement. We have changed “tumour core” to tumour cell compartment and have cited data that demonstrate hypo-vascularisation found in the juxta-tumoural stroma, due to PDAC cells inhibiting angiogenesis (PMID: 27288147). This paper supports our hypothesis of distribution of oxygen being reduced in the tumour area. Hence why we hypothesise that purinergic genes would be preferentially expressed in the tumour area: Line 112 “We hypothesised that genes related to high hypoxia scores would be expressed preferentially in the tumour cell compartment, as PDAC cells inhibit angiogenesis, causing hypo-vascularisation in the juxta-tumoural stroma (Di Maggio et al., 2016).”

We would like to clarify that we do not believe that only the tumour core suffers from hypoxia, we hypothesise that there is more hypoxia in the tumour cell areas. Although there are no reports of only the tumour core suffering from hypoxia, there is evidence of the tumour epithelial region of the cancer having a greater range of hypoxia (1-39%) compared to the stromal (1-13%) (PMID: 26325106). Moreover, all our analyses point to most purinergic genes differentially expressed in patients with high hypoxic scores being also related to cancer cells and the tumour region. These bioinformatic results linking certain genes like *P2RY2* and *ADORA2B* with hypoxia are also supported in published work cited in the discussion (Line 354 and 356).

I would suggest that the authors rely on published subtyping of PDACpatient cohorts (Collisson et al., 2010; Bailey et al; Moffit et al., 2015; Chan-Sen-Yue, 2020)and correlate the expression of purinome genes with the QM/basal-like PDAC subtype that has been confirmed multiple times as the "bad predictor" and use those subtypes for correlation with purinome pathway members. In figure 1E is also shown that P2RY2 is high in expression in basal-like subtype.

We thank the reviewer for this suggestion and have included the subtyping of patients in the PAAD-TCGA cohort in Sup. Table 1 and added comments about the genes related to the different subtypes in the text: Line 88 “In the Bailey model, most genes were related to the Immunogenic subtype except for NT5E, ADORA2B, PANX1 and P2RY2, which related to Squamous (Bailey et al., 2016). Collisson molecular subtyping showed several purinergic genes associated mostly to quasimesenchymal and exocrine subtypes (Collisson et al., 2011). The Moffit subtypes were not strongly associated with purinergic genes except for ADA, NT5E, P2RY6, P2RY2 and PANX1 associated with the Basal subtype (Moffitt et al., 2015).” and Line 345 “Expression of most purinergic genes was associated predominantly with immune cells, immunogenic PDAC subtype and low hypoxia scores (Figure 1C, E). In contrast, expression of genes correlated with worse survival and hypoxia (*PANX1*, *NT5E*, *ADORA2B* and *P2RY2*) was associated with tumour cells and the squamous PDAC subtype, correlating with hypoxia, inflammation and worse prognosis (Bailey *et al.*, 2016).”

We did not include the subtyping of Chan-Sen-Yue, 2020, due to the similarities with Moffit and the lack of correlation of basal/classical types with purinergic signalling genes as many of them are not expressed in cancer cells.

Figure 2: In further course of the paper the authors elaborate on possible functions of P2RY2 in PDAC. Although the mRNA data is pretty elaborate, the RNA SCOPE ISH has been performed on only 3 (!) patient PDAC samples. To demonstrated the mRNA is really found in tumor and not in normal adjacent tissue or stroma, I would strongly suggest to increase the number of samples here. The authors should perhaps try to co-localize ISH signals with IF/IHC for some other cancer cell marker, e.g. PanCK or GATA6/KRT81 in human samples to differentiate basal-like from classical samples; if possible, I would even suggest to perform immunohistochemistry instead of RNA scope and confirm the presence of the receptor. If there is an issue with the antibody availability, please state so in the manuscript so that it is clear to the readers why mRNA expression is favored over protein.

We thank the reviewer for these suggestions.

RNAscope was used to verify our transcriptomic bioinformatic results of location of expression P2Y2 in the tumour from publicly available data of 60 pairs of laser microdissection of PDAC epithelial and stromal tissue and the PAAD TCGA deconvolution of 177 patients. We have experienced issues with RNAscope due to the RNA degradation in pancreatic tissue and other technical difficulties which unfortunately led to only having 3 samples showing staining with the positive control. All three successful samples showed P2Y2 expression located in cancer cells. The images presented show the location of P2Y2 RNA expression in the tumour region, which was the aim of the RNAscope experiment.

RNAscope only captures mRNA expression above a specific threshold, and we are aware that P2Y2 will be expressed in other cell types in the normal adjacent as seen in the deconvolution. We have now included in supplementary single cell RNAseq data of normal PDAC tissue to counteract this issue (Sup. Figure 2B).

We also cite a publication that has performed P2Y2 IHC in 264 patients and showed that P2Y2 protein expression was predominantly shown in the epithelial tumour region (PMID: 30420446), hence staining of P2Y2 in a high number of patients has already been performed: Line 359 “These observations were supported by published immunohistochemical staining of 264 human PDAC samples, showing that P2Y_2_ localised predominantly in cancer cells in human PDAC”

As shown in Figure 1 E, P2Y2 is associated with basal and classical tumour cells, not just exclusively to basal, hence the staining to differentiate subtypes is not pertinent to the focus of this paper.

The GSEA data indicated that high P2Y2 expression relates to processes of adhesion/ECM/cytoskeleton organization where the authors draw the conclusion (based also on published data mostly on neuronal/astrocyte work) that P2Y2 may interact with integrins over the RGD domain and thus contribute to invasion an migration. Since this is a very important assumption, I would strongly suggest to expand the experiments of figure 2E and 2G on at least 2 more PDAC cell line, if possible include some with originally epithelial morphology (eg. HPAFII, HPAC…).The visualization of filaments can be done with common IF staining, eg. phalloidin, no need for stable expression.

Perhaps the reviewer missed Sup. Figure 2F, where data from Figure 2G are recapitulated in 3 different cell lines. We support the idea of the reviewer in including epithelial morphology cells, hence we added an extra cell line to have 2 cells with epithelial morphology, BxPC-3 and CAPAN-2.

We have tried repeating the experiment in Figure 2E in epithelial cells, but the way the epithelial cells grow in clusters (Sup. Figure 2F) make it very difficult to evaluate the morphology of individual cells and get quantifiable results. Nonetheless, we show phenotypic similarities of BxPC-3 to AsPC-1 cells in the invasion assays.

I would also be in favor of investigating the expression of EMT markers upon ATP stimulation.

We thank the reviewer for the suggestion, although feel this is out of scope for our study. There have been recent controversies with reference to EMT and cancer metastasis (PMID:31666716) but more importantly we see changes in cell morphology 1 hour after ATP treatment, indicating it is not/not just EMT.

How was 100µM/5µM chosen as a working concentration?

We have now included figures showing different concentrations of ATP (Sup. Figure 2D) and AR-C (Sup. Figure 2E) to illustrate how the concentrations were selected based on the greatest change in morphology for ATP and the full recovery of original cell morphology for AR-C.

AsPC-1 is also known as the cell line that gladly migrates and invades, usually used in metastatic modeling of PDAC. Would be interesting to see if another cell line that is not that migrative (HPAF II) presents the same effect…

This is an interesting point, although we haven’t performed experiments with low migrative cells, later on the work, invasion assays with the epithelial cell line BxPC-3, which has a very different migrative nature, presented the same effect (Sup. Figure 3G, F). We also perform invasion assays with PANC-1 cells, which also recapitulate an invasive phenotype when transfected with P2Y2.

Is treatment with ATP inducing expression of P2RY2 maybe? What is happening with Intergrin expression upon ATP treatment? Since the hypothesis is that extracellular ATP is driving the invasion, I would certainly suggest to investigate if ATP treatment induces expression of P2RY2 in a time and dose dependent manner.

We thank the reviewer for this suggestion. We have now changed the title to “Purinergic GPCR-integrin interactions drive pancreatic cancer cell invasion”, hence shifting from a focus on extracellular ATP and focusing on the effects of the RGD motif in invasion.

Figure 3:The authors made very good efforts here to provide functional evidence that P2Y2 is really involved and essential for ATP induced invasion in PDAC cells. They performed an 3D hanging drop spheroid model for invasion in co-culture with stellate cells and show that ATP treatment leads to invasive behavior that is than blocked by addition of P2Y2 antagonist or RGD blocking peptides. Although stellate cells are a nice add-on, keeping in mind the very complex tissue micro-environment of the PDAC, I don't rate the presence of stellate cells here as essential. Are the results the same when experiments are performed without stellate cells?

We thank the reviewer for raising this point, as it has allowed us to clarify that the stellate cells are crucial for this assay to work as they are essential for the formation of the cancer spheres due to their matrix deposition. We have included the hanging drop with and without stellate cells to illustrate this point (Sup. Figure 3A)

EMT markers increase upon ATP stimulation, do not increase under siRNA downregulation of P2Y2?

As mentioned above, we thank the reviewer for the comment, but we are not focusing on EMT, given the rapidity of the phenotype we observe.

Furthermore, the authors downregulate the P2Y2 using the siRNA/CRISPR-Cas9 approach and confirm that the P2Y2 is really involved in the invasive spread also using the specific RGD block. Experiments in the figure 3 are fairly done and provide functional evidence for the hypothesis. I would suggest that for clarity reasons on every panel (A, B,C…) is written which cell line is used (mostly Aspc1) and for the siRNA experiment I would suggest writing directly on the figure the time points (48h-72h post transfection) and shortly explain in the text why was mRNA evaluated as the measure of siRNA efficacy and not the protein? Probably the antibody problem, though western-blot applicable antibodies are available.

We thank the reviewer for acknowledging that the experiments in figure 3 provide functional evidence for our hypothesis. We agree with the reviewer and for clarity have included the cell line in each panel and the time point post transfection. We now include a Western blot showing protein levels in the siRNA P2Y2 treatment (Sup. Figure 3I).

Furthermore, for providing higher impact, I would encourage the experiments to be performed (at least in part) in a PDAC cell line with epithelial morphology (eg. HPAF II or any other that expresses the P2Y2 to a reasonable level).

We agree that performing this experiment with an epithelial morphology cell line provides higher impact, hence why we performed the experiment in BxPC-3 cell lines, perhaps missed in Sup. Figure 3G and H. We now highlight that they are epithelial-like in the text.

Figure 5: By using the DNA-PAINT method, the authors demonstrated that integrin av and P2Y2 physically interact in the cell membrane over the RGD domain and these interactions are essential for ATP induced P2Y2 mediated invasion in Aspc1 cells. The performed work seems plausible, however, I leave the technical evaluation of this experiment to experts in the field.General suggestion:I believe the work would benefit from a clinical/patient perspective if the authors show by immunohistochemistry in PDAC tissue samples that P2Y2 is localized at the invasive front/or metastasis. Is there a surrogate marker that can be used to label ATP rich regions in the tumor, are those regions at the invasive front? Are the P2Y2 positive cells those cells at the invasive front?

This is an interesting suggestion but immunostaining has already been performed on a large cohort of 264 PDAC patients (PMID: 30420446) and expression was consistent throughout the tumour cells.